# Diversity-Enhanced and Classification-Aware Prompt Learning for Few-Shot Learning via Stable Diffusion

## Abstract

Recent text-to-image generative models have exhibited an impressive ability to generate fairly realistic images from some text prompts. In this work, we explore to leverage off-the-shelf text-to-image generative models to train non-specific downstream few-shot classification model architectures using synthetic dataset to classify real images. Current approaches use hand-crafted or model-generated text prompts of text-to-image generative models to generate desired synthetic images, however, they have limited capability of generating diverse images. Especially, their synthetic datasets have relatively limited relevance to the downstream classification tasks. This makes them fairly hard to guarantee training models from synthetic images are efficient in practice. To address this issue, we propose a method capable of adaptively learning proper text prompts for the off-the-shelf diffusion model to generate diverse and classification-aware synthetic images. Our approach shows notable improvements in various classification datasets, with results comparable to existing prompt designing methods. We find that replacing data generation strategy of existing zero/few-shot methods with proposed method could consistently improve downstream classification performance across different network architectures, demonstrating its model-agnostic characteristic for few-shot learning. This makes it possible to train an efficient downstream few-shot learning model from synthetic images generated by proposed method for real problems.

## 1 Introduction

Recently, deep learning powered by large-scale annotated data has achieved great success in the field of image recognition [17]. However, acquiring and curating a large-scale high-quality dataset can be notoriously costly and time-consuming. This is especially true for inherently expensive domains, such as medical imaging, remote sensing, etc. Few-shot learning addresses the data issue by training a model using few data from the concerned tasks [87; 72; 63]. Generally, few-shot learning models use specialised algorithms and architectures to achieve the objective [99; 60; 105; 3; 98; 67]. This limits the variety of model architectures and potential applicability for real-world problems.

An alternative approach is to generate a synthetic dataset which is then used to train a classification model. In the early period, some efforts [5; 103; 26] explored the use of GANs for data generation in image recognition. However, constrained by the limited generative capabilities of early GAN models, the synthetic datasets usually address tasks on a small scale or only for a specific setting. Recently, text-to-image foundation generative models, e.g., DALL-E [51], GLIDE [47], Imagen [56], and Stable Diffusion [54], which are trained on billions of image-text pairs from web-datasets, have demonstrated impressive breakthroughs in generating high-quality images from text descriptions. It is hopeful not only to generate high-quality labeled data, but also achieve domain customization to train a classifier model tailored for the concerned tasks.

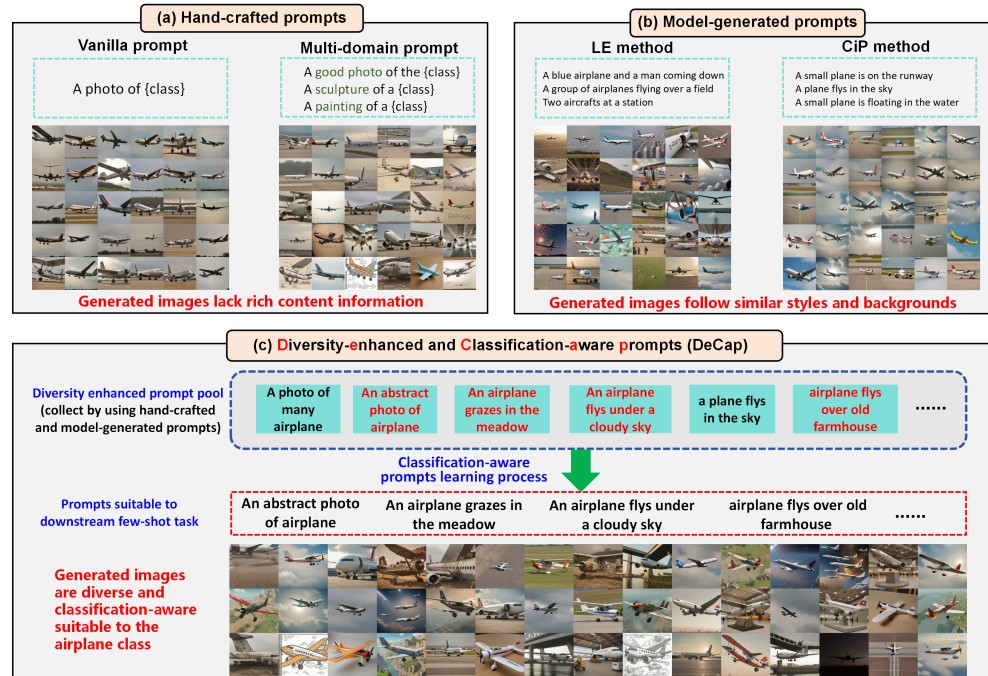

Figure 1: The comparison between existing prompt designing methods and proposed DeCap method. Hand-crafted methods usually generate images with different domain information but limited content information. Model-generated methods overcome this shortcoming, while may generate images share similar patterns. DeCap constructs a diversity-enhanced prompt pool by integrating the advantages of hand-crafted and model-generated methods, and then carry out classification-aware prompt learning process to mine proper prompts suitable to downstream few-shot tasks. Figure shows the mined prompts for airplane classification.

To achieve the goal, some researchers pay attention to designing proper text descriptions (prompts) of text-to-image generation models to generate desired synthetic images. A direct approach is to construct prompts by formatting class labels according to a template (called vanilla prompt [57; 50]), such as "a photo of {class}". To produce more diverse text descriptions, multi-domain prompt [62] additionally provides a list of domains with the prompt, e.g., " a {domain} of a {class}", to construct a set of prompt templates, in which '{domain}' refers to drawing, painting, sketch, etc. However, these hand-crafted prompts have limited capacity of generating images with rich content information, which usually leads to inferior generalization performance when training downstream models. To improve the content quality of prompts, the language enhancement (LE) method [18] leverages an off-the-shelf word-to-sentence T5 model to automatically expand class names into various sentences with rich content descriptions, containing the class names as language prompts. While this method hardly considers the class-relevant visual information for classification. The CiP method [37] generates high-quality prompts via extracting meaningful captions from real images using the off-the-shelf image captioning models such as BLIP2 [40], showing a significant improvement in generating informative synthetic images for better classification performance.

Although prompts produced by off-the-shelf foundational models can help generate high-quality images, they still have evident deficiencies in practice. On the one hand, generated prompts tend to share fixed or similar patterns for different images as reported in [82], which may limit diversity of synthetic images. For example, as shown in Figure 1, images generated by LE and CiP methods usually follow the similar styles and backgrounds. This limitation, which is even more serious under few-shot setting, may cause subpopulation shift problem [45; 92], i.e., some subpopulations of synthetic images shift from real-world datasets. On the other hand, existing prompt designing methods have relatively limited relevance to the

downstream classification tasks. Generally, the generated text prompts only employ class names or class-relevant visual information, which leads to some noises in generated prompts, e.g., prompts containing noisy labels or additional negative class information (please also see Figure 3). Therefore, it is relatively hard to guarantee that training models from synthetic images are efficient for downstream classification tasks, which tends to hinder their application effectiveness and reduce their performance stability in real problems.

To alleviate the aforementioned issues, this paper presents a **D**iversity-**e**nhanced and **C**lassification-**a**ware **p**rompt (**DeCap**) learning strategy to mine proper text prompts for downstream few-shot classification tasks (see Figure 1 for illustration). Our main idea is to combine existing hand-crafted diverse prompt templates and rich content prompt descriptions generated by off-the-shelf foundational models to construct a prompt pool containing potentially all-inclusive diverse prompt information. And then we propose a novel meta-learning approach to learn proper prompts tailored for the few-shot learning task. The DeCap method involves two nested learning loops: an inner-loop to train a classification model using generated synthetic images, and an outer-loop to search suitable prompts for text-to-image foundational generative models that produce synthetic training data for the inner-level classification model. The few-shot images are employed to compute outer-loop meta-objective for helping achieve classification-aware prompt learning. Through iteratively ameliorating both prompts selection and classification model performance, our algorithm is capable of mining proper prompts which are attained specifically suitable to concerned few-shot learning task.

In summary, this paper makes the following three-fold contributions:

(1) We proposed to automatically learn proper text prompts for text-to-image generative models to generate diverse and classification-aware synthetic images for few-shot learning task in a meta-learning manner.

(2) We verify that improving the diversity and classification-awareness of synthetic images could bring better downstream few-shot classification performance compared with existing prompt designing methods.

(3) We show that replacing data generation strategy of existing zero/few-shot methods could further improve downstream classification performance across different algorithms and network architectures.

The paper is organized as follows. Section 2 discusses related work. Section 3 presents the proposed method. Section 4 demonstrates experimental results and the conclusion is finally made.

## 2 RELATED WORK

**Text-to-Image Diffusion Model.** Diffusion model [21; 71] has emerged as a research hotspot in the field of image generation recently, due to their impressive generative capabilities. It achieves gradual matching from a Gaussian distribution to an image distribution by reversing the diffusion process. Recently, thanks to large-scale image-text paired datasets [59] and the maturity of text-image foundation models such as CLIP, some state-of-the-art text-to-image diffusion models, including DALL-E [51], GLIDE [47], Imagen [56], and Stable Diffusion [54], can produce a wide variety of highly realistic images, which has greatly propelled research in fields such as art [81], style transfer [95; 102], image controlling [55; 100; 14], data augmentation [77; 10] etc. In this paper, we explore leveraging off-the-shelf diffusion models to generate high-quality synthetic images for downstream few-shot image recognition.

**Synthetic Dataset for Image Recognition.** In the early stages, some research[5; 103; 26] explored the role of synthetic datasets with GAN models. However, due to the limited data generation capabilities of early GANs, the application scenarios are significantly constrained. With the emergence of large-scale text-to-image generative models, recent studies have validated the utility of synthetic datasets at a large scale. For example, for classification tasks, [57; 4] train synthetic ImageNet datasets from scratch, [18; 38] showing that CLIP [50] can boost performance from synthetic datasets. [76] validates the outstanding performance of synthetic dataset using SimCLR and MAE models. In the field of object detection, [29] utilizes the output results of generative model's cross-attention layers as weak supervision for zero-shot object recognition. Additionally, synthetic datasets are also applied to addressing long-tail problems [61].

The data generation strategy could be roughly divided into two categories. One is fine-tuning based method [2; 96], which fine-tunes generative models' parameters using task data. These methods demonstrate strong domain adaptation capabilities on large-scale datasets and can effectively generate samples that conform to the distribution of real dataset. However, it often requires large-scale real datasets. Therefore, the other is prompt designing method to address few-shot learning. They don't alter the parameters of generative models; instead, it focuses on setting proper prompts for off-the-shelf generative models to generate synthetic datasets. As discussed in Section 1, there exist two methodologies of setting prompts, i.e., hand-crafted and model-generated prompts. While they are not sufficient to generate high-quality images for classification. In this paper, we propose to integrate the advantages of both methodologies to achieve a diversity-enhanced and classification-aware prompt learning strategy. We need to clarify that, different from prompt learning methods [106; 107] specifically designed for multimodal models like CLIP, which directly helps adjust off-the-shell models prediction adapting to the concerned data, our prompt learning strategy focuses on generating efficient synthetic data for further help train downstream few-shot learning.

**Meta Learning.** Meta learning[22; 65], also known as learning to learn, focuses on how to quickly adapt and apply previously acquired knowledge when faced with new learning tasks. Meta learning is widely used in few-shot learning [12; 63; 52; 69], hyperparameter optimization [13], transfer learning [27; 74], label noise learning [64; 66; 89], machine learning automation [90], etc. For image generation field, meta learning is used to achieve data distillation [46; 86; 85; 73], data augmentation [91], etc. Different from previous works updating parameters of generative model, we use meta learning technique to learn proper text prompts of generative models to generate high-quality synthetic images for concerned few-shot learning task.

## 3 THE PROPOSED DeCap METHOD

### 3.1 PRELIMINARY

For a $N$-classification task, We use $\hat{x}_{ij}^{(k)} = g(\theta_{ij}, \epsilon_k)$ to denote the generated image $\hat{x}_{ij}^{(k)}$ via an off-the-shelf text-to-image foundational models $g$, where $i \in [N], [N] = \{1, \cdots, N\}$ represents the $i$-th class, $j \in [M], [M] = \{1, \cdots, M\}$, where $M$ means how many different prompts for this class, $\theta_{ij}$ represents the prompt used to generate this image, $\epsilon_k$ represents random gaussian noise. We denote the mini dataset generated by prompt $\theta_{ij}$ as $X_{ij}^{syn} = \{\hat{x}_{ij}^{(k)}, k = 1, 2, \cdots, l\}$, where $l$ means the generation number of each prompt. We only study prompt setting for image generation, and we will drop explicit dependence of $X_{ij}^{syn}$ on $\epsilon_k$ for brevity in the following, i.e., $X_{ij}^{syn} = g(\theta_{ij})$. Our approach can be directly applied to different diffusion models, and in this work we study the open-sourced model: Stable Diffusion (SD) [54].

Considering a few-shot classification task with real data $D^{real} = \{(x_{ij}, y_{ij}), i = 1, \cdots, N, j = 1, \cdots, K\}$, where $x_{ij}, y_{ij}$ denote image and its label, and $N, K$ denote the number of classes and samples of each class, respectively. To boost few-shot model performance, it could use SD model to help generate high-quality synthetic data for few-shot image recognition tasks. Specifically, the synthetic data could be formulated as

$$X^{syn} = g(\boldsymbol{\theta}), \boldsymbol{\theta} = \{\boldsymbol{\theta_i}, i \in [N]\}, \boldsymbol{\theta_i} = \{\theta_{i1}, \theta_{i2}, \cdots, \theta_{iM}\}, X^{syn} = \{X_{ij}^{syn} = g(\theta_{ij}), i \in [N], j \in [M]\}.$$

For simplicity, we denote the obtained synthetic data as $D^{syn}(\boldsymbol{\theta}) = \{X^{syn}(\boldsymbol{\theta}), Y\}$, where $Y = \{Y_i, i \in [N]\}, Y_i = \{y_{i1}, \cdots, y_{iM}\}$. Based on $D^{syn}(\boldsymbol{\theta})$, we could train a classification network $f_w$ by optimizing the following objective:

$$w^* = \arg \min_{w \in \mathcal{W}} \mathcal{L}^{task}(f_w, D^{syn}(\boldsymbol{\theta})), \tag{1}$$

where $\mathcal{W}$ denotes parameter space, $\mathcal{L}^{task}(f_w, D^{syn}(\boldsymbol{\theta})) = \frac{1}{MN} \sum_{i=1}^{N} \sum_{j=1}^{M} \mathcal{L}^{task}(f_w(x_{ij}), y_{ij})$, and $\mathcal{L}^{task}$ denote the classification loss for the downstream few-shot learning task, e.g., cross-entropy loss.

As discussed in Section 1, existing prompt designing methods may generate limited diversity of synthetic images, tending to degrade generalization performance when training downstream classification models.

Especially, we could see that the prompt construction process of existing methods has limited relevance to the downstream classification tasks from Eq.(1), i.e., drops the explicit dependence of $w^*$ on $\boldsymbol{\theta}$. In other words, existing prompt learning methods are classification-agnostic, which greatly reduces the alignment between synthetic datasets and downstream classification task requirement. To address these two issues, we propose a novel prompt learning strategy called DeCap, which explores to learn proper prompts for generating high-quality images to improve downstream few-shot learning task. We present the method and solving algorithm in Section 3.2 and 3.3, respectively.

### 3.2 PROPOSED DeCap Method

The proposed DeCap method firstly constructs a diversity-enhanced prompt pool (Section 3.2.1) by integrating the advantages of hand-crafted and model-generated methods, and then carry out classification-aware prompt learning process (Section 3.2.2) to mine proper prompts suitable to downstream few-shot task.

#### 3.2.1 DIVERSITY-ENHANCED PROMPT POOL CONSTRUCTION

In this section, we proposed to integrate the advantages of both hand-crafted and model-generated methods to construct a prompt pool that contains potentially all-inclusive diverse prompt information.

Specifically, we construct a unique prompt pool $\Theta$, which contains hand-crafted prompts and model-generated prompts, for every class in the dataset. For hand-crafted prompts, we first select some common prompt templates provided by [50] which contain various domain information. Then we manually add some new prompts into the pool, covering aspects such as color, style, camera angle and so on. Since these prompts describe the object in general terms, we share these prompts for all classes. For model generated prompts, we use BLIP2 model as CiP method [37] to describe images from few-shot datasets, and utilize T5 model as LE method [18] to generate corresponding class prompts with class labels as information. These prompts describe the object in detail, so different classes will have totally different descriptions. In conclusion, for each category's prompt $\boldsymbol{\theta}_i$, it consists of two parts: the hand-crafted prompt $\boldsymbol{\theta}_i^h$ and the model-generated prompt $\boldsymbol{\theta}_i^m$, i.e., $\boldsymbol{\theta}_i = [\boldsymbol{\theta}_i^h, \boldsymbol{\theta}_i^m]$, where all classes share the same template $\boldsymbol{\theta}_i^h$, while possess private prompt $\boldsymbol{\theta}_i^m$.

After conducting this process, there already exists adequate prompts containing both diverse domain and content information in the prompt pool. However, this prompt pool is overly abundant and classification-agnostic, which contains not only proper prompts but also noisy prompts for downstream few-shot learning task. An illustration of the necessity of using adaptive prompt learning please see Appendix D.1. Therefore, we further propose a classification-aware prompt learning strategy to mine proper prompts form the prompt pool in a meta-learning manner to help generate high-quality images suitable for downstream few-shot task. We give a simple example about what our prompt pool looks like in Appendix B.1.

#### 3.2.2 CLASSIFICATION-AWARE PROMPT LEARNING

The main idea is to establish the direct connection between prompt setting process and downstream classification model learning. Inspired by recent meta learning methods [65; 73; 22], we formulate the classification-aware prompt learning as the following bi-level optimization objective:

$$\boldsymbol{\theta}^* = \arg \min_{\boldsymbol{\theta} \in \Theta} \mathcal{L}^{meta}(f_{w^*(\boldsymbol{\theta})}, D^{real}), \tag{2}$$

$$\text{where } w^*(\boldsymbol{\theta}) = \arg \min_{w \in \mathcal{W}} \mathcal{L}^{task}(f_w, D^{syn}(\boldsymbol{\theta})), \tag{3}$$

where the inner-level objective (Eq.(3)) is the same as Eq.(1), while we explicitly require the performance of classification model to depend on the prompts $\boldsymbol{\theta}$. Specifically, given a prompt set $\boldsymbol{\theta} \in \Theta$, we use these prompts to obtain the synthetic dataset $D^{syn}(\boldsymbol{\theta})$, and then train the downstream classification model on the synthetic dataset. Different from existing method preassigning the prompts, we want to learn proper prompts to generate high-quality data that more suitable to downstream task. To this goal, we use few-shot data $D^{real}$ given by the downstream tasks to compute the outer-level meta loss $\mathcal{L}^{meta}$ for evaluating the performance

Figure 2: **Overview of the proposed DeCap method**. DeCap training involves two nested training loops. In the inner-loop optimization, we use the selected prompts set $\theta$ to generate synthetic dataset and then help train a downstream classification model, while in the outer-loop optimization, we search proper prompts from pre-constructed prompt pool which are attained specifically suitable to few-shot learning task.

---

**Algorithm 1** Learning Algorithm of the Proposed DeCap Method

---

**Input:** Downstream few-shot learning task dataset $D^{real}$; Algorithm iteration number *max-iter*, population quantity *popsize*; Prompt pool *pool*; off-the-shell text-to-image generative model $g$

**Output:** Optimal prompt set $\theta^*$

---

1: GA.initial(max-iter,popsize)                                  ▷ Genetic Algorithm (GA) initialize
2: **for** $iter = 1, 2, \ldots, max\text{-}iter$ **do**
3:     fitness=[] , Pop=[]
4:     **for** $m = 1, 2, \ldots, popsize$ **do**
5:         $pop^{(m)}$ =GA.sample()                              ▷ An individual in the population
6:         $\theta^{(m)}$ , $Y^{syn}$ = get_prompt($pop^{(m)}, pool$)   ▷ See Algorithm 2 in Appendix B.2
7:         $X^{syn}(\theta^{(m)}) = g(\theta^{(m)})$
8:         $w^*(\theta^{(m)}) = \arg\min \mathcal{L}^{task}(f_w, D^{syn}(\theta^{(m)}))$   ▷ Inner-loop Optimization
9:         $fitness_{pop^{(m)}} = \mathcal{L}^{meta}(f_{w^*(\theta^{(m)})}, D^{real})$   ▷ Outer-loop Optimization
10:         fitness.append($fitness_{pop^{(m)}}$) , Pop.append($pop^{(m)}$)
11:     **end for**
12:     GA.update(fitness,Pop)                                  ▷ Updating searching direction
13: **end for**
14: return GA.best

---

of obtained classification model in Eq.(2), so as to learn proper prompts $\theta$. Through iteratively ameliorating both searching prompts at outer-level learning and classification model performance at inner-level learning, our algorithm is capable of mining classification-aware prompts which is attained specifically suitable to downstream few-shot learning task. In our implementation, the optimization of $\theta \in \Theta$ is actually a discrete prompt selection problem. We will introduce the solving algorithm in the next section.

### 3.3 LEARNING ALGORITHM OF THE PROPOSED DECAP METHOD

Considering that the optimization of prompt $\theta$ is a discrete search problem, we use the genetic algorithm (GA) [31] to solve the outer-level optimization objective in Eq.(2). Generally speaking, a genetic algorithm first generates different inputs, then obtains the corresponding value function outputs for these inputs, adjusts the search direction based on the magnitude of the outputs, and eventually completes the optimization process. Therefore, we only need to define the GA's input and value function for DeCap objective, and then genetic algorithm can be employed to mine proper prompts $\theta^*$ from prompts pool $\Theta$.

Table 1: Top-1 accuracy on different datasets. **Bold scores** represent the best result on each dataset, and the second best scores are marked by orange.

| | STL-10 | CIFAR10 | Im-10 | Pets | Caltech-101 | Im-100 | EuroSAT | Aircraft | Country211 |
|---|---|---|---|---|---|---|---|---|---|
| *without real* | | | | | | | | | |
| zero-shot | 94.26 | 70.25 | 97.22 | 81.85 | 83.89 | 70.14 | 23.11 | 17.07 | 13.44 |
| vanilla prompt | 95.33 | 72.37 | 97.69 | 82.29 | 84.74 | 70.62 | 31.31 | 17.04 | 13.72 |
| multi-domain | 94.97 | 70.66 | 97.89 | 83.07 | 87.56 | 70.50 | 30.11 | 17.85 | 13.90 |
| LE | 94.61 | 70.33 | 97.45 | 83.24 | 84.03 | 70.73 | 29.35 | 17.73 | 14.14 |
| CiP | 94.92 | 70.24 | 97.65 | 84.04 | 88.12 | 70.76 | 39.91 | 18.00 | 14.98 |
| DeCap (ours) | **95.91** | **76.98** | **97.95** | **85.36** | **88.67** | **71.08** | **41.94** | **19.74** | **15.44** |
| *with real* | | | | | | | | | |
| real-only | 94.28 | 70.33 | 97.22 | 81.96 | 84.51 | 70.33 | 24.15 | 19.41 | 13.80 |
| vanilla prompt | 95.55 | 76.20 | 98.00 | 83.84 | 89.85 | 70.87 | 47.83 | 18.21 | 13.77 |
| multi-domain | 95.02 | 74.54 | 97.92 | 84.56 | **90.31** | 70.62 | 43.30 | 18.99 | 13.90 |
| LE | 94.72 | 71.66 | 97.49 | 84.00 | 84.34 | 70.46 | 42.06 | 20.22 | 14.37 |
| CiP | 95.05 | 70.51 | 97.75 | 85.16 | 89.86 | 70.86 | 49.17 | 20.31 | 15.49 |
| DeCap (ours) | **95.93** | **77.19** | **98.03** | **85.78** | 89.87 | **71.11** | **50.22** | **20.64** | **15.68** |

In our problem, the input is defined as a vector of integers. The length of the vector represents the number of prompts selected, and each dimension of the vector corresponds to the index of the selected prompt, with values ranging from 0 to the size of the prompt pool. Under this definition, each input represents a different combination of selected prompts. The value function is defined as the outer-level meta loss $\mathcal{L}^{meta}$ in Eq.(2).

For each category, the prompt $\boldsymbol{\theta}_i$ includes the same hand-crafted prompts $\boldsymbol{\theta}_i^h$ shared for all categories and the class-specific model-generated prompt $\boldsymbol{\theta}_i^m$. This hypothesis could effectively reduce the number of parameters for setting prompts. We believe this configuration is reasonable because domain information could typically be shared, while class-specific content descriptions cannot. Our DeCap method is able to balance the common patterns across categories with the unique differences specific to each category. The whole learning algorithm of proposed DeCap method is summarized in Algorithm 1. More details about genetic algorithm please see Appendix B.3.

## 4 EXPERIMENTAL RESULTS

### 4.1 FEW-SHOT CLASSIFICATION PERFORMANCE

We compared with existing prompt designing strategies including: (1) vanilla prompt [50]: using the template "a photo of {class}". (2) multi-domain prompt: using different text templates from domains provided in [50]. (3) LE [18]: using the T5 model [1] for text prompt construction, where the input and output of T5 model are the class label and a sentence containing the class label, respectively. (4) CiP[37]: generating captions for real image data using the BLIP2[2] model. We conduct experiments on 9 datasets: CIFAR10[34], STL-10[7], Imagenette[24](Im-10), Pets[48], Caltech-101[11], ImageNet100[75](Im-100), EuroSAT[19], FGVC Aircraft[43] and Country211[50]. Datasets details are introduced in Appendix C.1. For the selection of the classification model, we use the CLIP model, as it has shown powerful classification ability. The training strategy we used strictly follows the settings described in [18], where we finetune CLIP with generated data. We use "a photo of {class}" as the text initialization for CLIP tuning for all datasets to eliminate the impact of different initializations on the evaluation of each method. Training and evaluating details are presented in Appendix C.2. Table 1 shows the few-shot classification performance of each method on six downstream few-shot learning datasets, where "without real" means that we only use synthetic datasets to train downstream models, while "with real" means that we use both synthetic and real few-shot images to train downstream models. Some ablation studies on DeCap method please see Appendix D.2.

Using synthetic data to train downstream classification model, DeCap method demonstrates the best classification accuracies across diverse datasets. All prompt designing methods can improve CLIP zero-shot per-

---

[1]https://huggingface.co/mrm8488/t5-base-finetuned-common_gen
[2]https://huggingface.co/Salesforce/blip2-opt-2.7b

Table 2: Comparion of DeCap and SOTA methods on different datasets. "Method + DeCap" denotes the performance of replacing original synthetic data strategies of each method with DeCap method.

| | STL-10 | CIFAR10 | Im-10 | Pets | Caltech-101 | Im-100 | EuroSAT | Aircraft | Country211 | avergae |
|---|---|---|---|---|---|---|---|---|---|---|
| FakeIt[57] | 52.26 | 38.45 | 69.60 | 29.74 | 66.20 | 32.75 | 48.40 | 37.70 | 3.61 | 42.08 |
| With DeCap | **60.39** | **48.80** | **75.40** | **55.22** | **70.51** | **39.21** | **51.20** | **40.60** | 4.22 | **49.51** |
| SuS-X[78] | 95.24 | 72.77 | 98.24 | 79.64 | 84.57 | 69.96 | 33.89 | 18.30 | 12.96 | 62.84 |
| With DeCap | **95.43** | **75.89** | **98.39** | **80.40** | **84.89** | **70.3** | 37.37 | 19.83 | **13.02** | **63.94** |
| CaFo[101] | 95.33 | 85.34 | 97.66 | 86.62 | 94.09 | 74.64 | 83.5 | 26.07 | 16.20 | 73.28 |
| With DeCap | **95.90** | **86.00** | **98.06** | **88.66** | **94.28** | **76.28** | **84.46** | **26.76** | 16.88 | **74.14** |

formance, showing that generating synthetic data is helpful to train downstream classification model. As for datasets with simple categories like STL-10, CIFAR-10 and Im-10, the hand-crafted prompts could achieve superior performance than model-generated prompts, illustrating that the prompts with only class/domain information may be relatively more proper for these tasks; while for datasets with complex categories like Pets, Caltech-101 and Im-100, the model-generated prompts could achieve better performance than hand-crafted prompts, implying that rich content information is more helpful to address these complex tasks. These results reveal that effective prompts should be set based on concerned task information. To this goal, proposed DeCap method could adaptively learn proper prompts suitable to the concerned tasks by reconciling class/domain information and rich content information (visualization of mined prompts see Appendix E.2), so as to achieve an average performance improvement of 1.30% point compared to the best results of existing method on different datasets. We also evaluate the adversarial robustness of these methods in Appendix D.4, which further substantiate the high-quality data generation capability of DeCap method.

When using additional real data, CLIP's performance could be further improved, though the number of real data is relatively smaller than synthetic data. This implies that the quality of real data may be higher than that of synthetic data. All prompt designing methods obtain a further improvement over only using synthetic data. Even so, DeCap method still shows advantages over other methods on most datasets, demonstrating that our approach could genuinely augment few-shot datasets. These experimental results support the capability of proposed DeCap method in generating high-quality images for downstream few-shot learning tasks.

## 4.2 COMPARION WITH SOTA METHODS

In Section 4.1, we showed that under the same CLIP model architecture, DeCap performs well compared with other prompt designing methods. The key goal of DeCap method is to mine proper prompts to generate high-quality data for downstream few-shot learning, while it is not confined to specialised algorithms and architectures to complete few-shot learning tasks. To illustrate this, we explore to use synthetic data of DeCap method to evaluate its performance on other zero/few-shot algorithms and architectures.

Specifically, we conducted our experiments on three SOTA algorithms: (1) FakeIt [57]: It uses synthetic datasets to train the network on ResNet-50. (2) SuS-X [78]: It leverages synthetic datasets as a dynamic support set and extends Tip-Adapter by utilizing the image-text distance. (3) CaFo [101]: It augments few-shot datasets with synthetic data and then combines the predictions of pre-trained CLIP and DINO. In our implementations, we replaced the data generation strategies of these methods with DeCap without altering any of model architectures for a fair comparison, and follow original settings of these methods to train the corresponding classification models. More details please refer to Appendix C.3.

Table 2 reports the results. Notice that FakeIt method uses synthetic data to train the ResNet-50 model from scratch, which eliminates effects of pre-training data for downstream tasks. Thus performance of the trained classification model could appropriately reflect the quality of synthetic data. The DeCap method achieves a significant improvement of 7.43% point over original data generation strategy of FakeIt, substantiating the capability of our method in generating high-quality data suitable to concerned tasks. Though SuS-X and CaFo methods use pre-trained models, synthetic data of DeCap method could still outperform these methods in the vast majority of datasets. These results demonstrate that synthetic data of our DeCap method are not confined to specialised algorithms and architectures. This implies that our DeCap method is model-agnostic for downstream few-shot learning tasks, and hopeful to be readily applied to real-world problems and tasks.

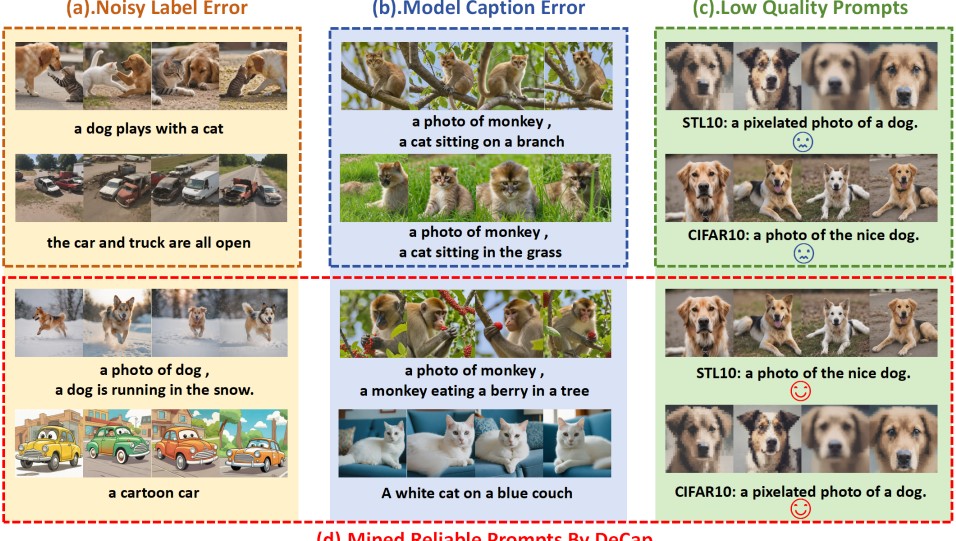

Figure 3: Illustration of (a) noisy label, (b) model caption error or (c) low quality prompts in prompt pool generated by existing prompt designing methods, and (d) mined reliable prompts by our DeCap method.

### 4.3 WHY PROPOSED DECAP METHOD PERFORM BETTER?

In this section, we further present some analysis of DeCap method in two aspects: robustness against noisy or low quality prompts, and data value analysis of synthetic data.

#### 4.3.1 ROBUSTNESS AGAINST NOISY OR LOW QUALITY PROMPTS

Existing prompts methods may set noisy or low quality prompts for downstream tasks. For LE method, it may generate prompts that contain not only the class we want, but also other classes in the dataset. An illustrated example is presented in Fig.3 (a): for STL10 dataset, when we generate images for "dog"/"car" classes, some images also contain information of "cat"/"truck" classes. Since "cat"/"truck" classes belong to the dataset, these prompts would generate images with noisy labels for the classification of "dog"/"car". For CiP method, due to the limitations of the BLIP2 model's capability, it cannot always accurately annotate images, which may result in misidentifications. Although CiP method recognizes this issue and employs a prompt concatenation method like "a photo of {class}, {image caption}" to reduce the influence of noisy captions, we found this may not always work. For example, as shown in Fig.3 (b), when the BLIP2 model mistakenly identifies a monkey as a cat, the defined prompt "a photo of monkey, a cat sitting in a branch" may generate an image that blending features of cat and monkey. The issue of misidentification is particularly prominent in certain tasks, such as CIFAR10, where the low resolution images significantly impact the model's judgments. This explains why the CiP method performs poorly on CIFAR10 dataset, as presented in Table 1. Moreover, hand-crafted prompts often introduce different domain information to construct diverse prompts. Generally, only part of domain information is reliable, while an amount of domain information may be of low quality for the concerned tasks. As shown in Fig.3 (c), though both of prompts could generate images of dog, the improper domain information could hinder the performance of concerned classification models, e.g., the synthetic pixelated images may provide low-quality training data for STL-10 task. In Appendix D.6,we further illustrate influence of prompts with domain information on the synthetic images.

Unfortunately, these noisy prompts are relatively hard to be filtered using data cleaning strategies such as CLIP filtering [18]. To address the issue, proposed DeCap method aims to mine proper prompts suitable to the concerned classification task in a meta-learning manner. As shown in Fig.3 (d), with such higher-level downstream classification-aware outer-loop supervised information, DeCap method could adaptively select effective prompts that help boost downstream classification performance, and discard aforementioned potential noisy prompts that would potentially hurt downstream classification performance.

| a doodle of the car. | dog walking on a sunny day. | a photo of the bird. | deer on a green pond. |
|---|---|---|---|
| | | | |
| olympic athletes racing cars during racing match. | a photo of dog , a dog is running in the snow | a embroidered bird. | a photo of the clean deer. |
| | | | |
| A dynamic car. | a photo of dog, a dog plays in water with stick | art of the bird. | a cartoon deer. |
| | | | |

Table 3: Examples of synthetic images generated by DeCap method for STL-10 dataset.

### 4.3.2 DATA VALUE ANALYSIS OF SYNTHETIC DATA

To better analyze why DeCap method outperforms existing prompt designing methods, we use "leave-one-out" method [16] to evaluate data valuation, and then select typical high-quality images generated by DeCap method. Specifically, given a dataset $S$ and a measure function $V$, we use $\phi_i = V(D \cup \{i\}) - V(D)$ to represent data valuation of the synthetic image $i$. In our implementation, we use the dataset generated by vanilla prompt method as the benchmark dataset $S$ and classification accuracy as the measure function $V$. Then we could compute data valuation of synthetic images generated by DeCap method via adding one image at a time. Table 3 visualizes the synthetic images with high data valuations for STL-10 dataset, and more visualizations are shown in Appendix E.1. As shown, we can see that synthetic images contain various patterns such as image style, background, camera angles, and actions, providing novel, diverse, and meaningful content information for original sparse data. This indicates that DeCap method does mine proper diverse and rich content prompts suitable to concerned downstream few-shot learning tasks, naturally leading to its better accuracy than other prompt designing methods.

## 5 CONCLUSION

We present the DeCap, a novel adaptive prompt learning approach to generate diverse and classification-aware synthetic data for downstream few-shot learning in a meta-learning manner. Proposed DeCap method could mine potential reliable prompts suitable to downstream few-shot learning tasks, demonstrating impressive capabilities in improving downstream classification models for different few-shot learning tasks compared with existing prompt designing methods. We could further boost existing SOTA zero/few-shot learning methods by simply replacing data generation strategy with the proposed method, showing its potential model-agnostic characteristics. Besides, we also provide some intuitive visual interpretation, providing an initial insight into proposed DeCap method. Such an adaptive prompt learning approach is hopeful to be employed to other computer vision tasks, like semantic segmentation and object detection, etc.

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

## A  Limitations

Although our DeCap method performs well among different datasets and model architectures compared with existing prompt designing methods, we have to admit that DeCap has the following limitations. Firstly, DeCap requires more training cost. It usually spends 160 GPU hours to mine proper prompts on small scale datasets such as STL-10 and CIFAR10, and for large scale datasets such as Imagenet100, the cost will go up to nearly 700 GPU hours. However, it is worth emphasizing that once we finish training, the prompts we have learned could be used to generate sufficient images for training other few-shot algorithms and model architectures. Secondly, the search space of prompt set for DeCap method is confined to the pre-constructed prompt pool, which may lead to suboptimal solutions. One potential strategy is to learn continuous soft prompts just like what [28; 107; 70] do. However, the computation of meta gradients for learning soft prompts requires unaffordable memory: even a Nvidia A800 GPU can't support the backward of a single synthetic image. Note that the suboptimal solutions of DeCap method could achieve impressive performance, we believe more advanced prompt learning strategy would further boost the downstream classification models. Lastly, compared with model-generated prompt methods, proposed DeCap method seems to lack extensibility. One promising idea is to learn a prompt generator that produces prompts conditioned on concerned tasks. We leave the above potential shortcomings for future work, and we also look forward to the emergence of following works to address these problems.

## B  More details of proposed DeCap Method

### B.1  Examples of prompt pool construction

In this section, we give a simple example about what our prompt pool looks like.

Let us consider "cat v.s. dog" classification task. Assuming that our hand-crafted prompts are ["a photo of {}", "a sketch of {}", "a {} image"] and model-generated prompts are {cat:["a cat on the grass", "a cute cat"], dog:["a barking dog", "a dog in the room"]}. Then, our prompt pool will be:

{cat:["a photo of {cat}", "a sketch of {cat}", "a {cat} image","a cat on the grass", "a cute cat"],
dog:["a photo of {dog}", "a sketch of {dog}", "a {dog} image","a barking dog", "a dog in the room"]}

If we randomly select 2 prompts for each class, for example, the 0th and 3th prompts for cat, and 1th and 2th prompts for dog, which represents $pop = [0, 3, 1, 2]$ , the selected prompts for generating dataset will be {cat:["a photo of {cat}","a cat on the grass"; dog:"a sketch of {dog}","a {dog} image"]}.

If we share hand-crafted prompts, for example, assuming we select the prompt template "a photo of {}", then it means that ["a photo of {cat}", "a photo of {dog}"] will be selected to help generate dataset.

### B.2  "get_prompt" method in Algorithm 1

Algorithm 2 shows the "get_prompt" method in Algorithm 1. We denote the number of classes as $N$, the name of these classes as "class_names", prompt numbers per class as $M$.

**Algorithm 2** Get_prompt Algorithm

**Input:** indexes $pop$, prompt pool $pool$; hyper-parameters including: whether share hand-crafted prompts $share$, hand-crafted prompts numbers $n$;
**Output:** prompt set: prompts, labels: $Y^{syn}$

---

1: pop.reshape[$N,M$]        $\triangleright$ pop is the index of $\boldsymbol{\theta} = [\boldsymbol{\theta}_1, \boldsymbol{\theta}_2, \cdots, \boldsymbol{\theta}_N]^\top, \boldsymbol{\theta}_i \in \mathbb{R}^M$ in prompt pool
2: **if** $share$ **then**
3:   pop[:, : $n$]=pop[0, : $n$].repeat[$N$,1]     $\triangleright$ we make all the first $n$ elements of $\boldsymbol{\theta}_i$ the same
4: **end if**
5: prompts=[], $Y^{syn}$=[]          $\triangleright$ $Y^{syn}$ contains every synthetic sample's label
6: **for** $i = 1, 2, \cdots N$ **do**
7:   class=class_names[$i$]
8:   prompts.append($pool$[class][pop[$i$]])
9:   $Y^{syn}$.append($i$.repeat[$M$])
10: **end for**
11: return prompts, $Y^{syn}$

---

## B.3 GA ALGORITHM DETAILS

Genetic Algorithm (GA) is an optimization technique inspired by natural selection and genetic processes, widely used for complex problem-solving. Its key steps can be summarized as follows:

- Initialization of Population: Randomly generate a set number of individuals (solutions) to form the initial population, with each individual represented by a gene encoding (typically a binary string or real numbers).

- Fitness Evaluation: Assess the fitness of each individual using a fitness function that quantifies their performance based on the problem's objectives.

- Selection: Select individuals for the next generation based on their fitness values. Common selection methods include roulette wheel selection, tournament selection, and rank selection, where fitter individuals have a higher chance of being chosen.

- Crossover: Combine parts of two parent individuals' genes to produce new offspring. Crossover enhances genetic diversity, with methods like single-point, multi-point, and uniform crossover.

- Mutation: Introduce random changes to a portion of an individual's genes with a certain probability, increasing genetic variation and helping to avoid local optima. Mutation can involve flipping gene bits or assigning random values.

- Population Update: Merge the offspring with the current population and select suitable individuals based on fitness, often using elitism to retain the best solutions.

- Termination Condition: Determine if termination criteria are met, such as reaching a maximum number of iterations, achieving a predefined fitness goal, or when improvements in fitness become negligible.

- Output Results: Present the final optimal solution or any satisfactory solutions, along with relevant analysis and validation.

Actually, in Algorithm 1, the *GA.update()* operation means the steps from "selection" to "population update" operation. Our code are based on the scikit-opt library, and we use their default operators. What's more, unlike traditional meta learning methods[12; 63; 13; 27; 74] relying on computing meta gradient to optimize outer-level meta loss, our outer-level optimization does not involve any meta gradient calculation (i.e., derivative-free optimization), and we only execute gradient descent algorithm at the inner-level optimization.

## C  Implementation Details

### C.1  Datasets Details

In this section, we give a brief introduction about datasets we used in Section 4.

**CIFAR10**: The CIFAR10 dataset contains 10 common classes: airplane, car, bird, cat, dog, deer, frog, horse, ship, truck. Each class contains 6000 color images with $32 \times 32$ size. CIFAR10 is widedly used in image classification.

**STL-10**: The STL-10 dataset contains 10 common classes in real life: airplane, bird, car, cat, deer, dog, horse, monkey, ship, and truck. Although these photos comes from ImageNet, their annotations may be quite different, for example, "dog" class contains various dog breeds.

**Imagenette**: Imagenette is a subset of the larger ImageNet dataset, containing 10 easily distinguished classes: tench, English springer, cassette player, chain saw, church, French horn, garbage truck, gas pump, golf ball, parachute. It was created to provide a smaller, more manageable subset for training and testing image classification models.

**Pets**: The Pets dataset consists of images of 12 different cats breeds and 25 different dogs breeds. It is commonly used for fine-grained classification tasks, where the goal is to classify images into specific sub-categories within a broader class.

**ImageNet100**: ImageNet100 is a subset of the original ImageNet dataset, containing 100 classes. It serves as a smaller alternative to the full ImageNet dataset for training and evaluating deep learning models for image classification tasks.

**Caltech-101**: The Caltech-101 dataset is a widely used benchmark dataset for object recognition. It contains images of objects belonging to 101 distinct categories, including animals, vehicles, and household items.

**EuroSAT**: EuroSAT is a dataset of Sentinel-2 satellite images for land cover classification. It contains 27,000 RGB images across 10 classes, such as agriculture, forest, and water bodies, with a resolution of 64x64 pixels. It is widely used in remote sensing and environmental monitoring tasks.

**Aircraft**: The FGVC Aircraft dataset is designed for fine-grained visual classification of aircraft. It includes 10,000 images of 102 different aircraft models, focusing on distinguishing subtle differences between similar models. It is commonly used in fine-grained recognition research.

**Country211**: Country211 is a dataset released by OpenAI, designed to assess the geolocation capability of visual representations. It filters the YFCC100m dataset to find 211 countries that have at least 300 photos with GPS coordinates. OpenAI built a balanced dataset with 211 categories, by sampling 200 photos for training and 100 photos for testing, for each country.

### C.2  Experiment Settings in Section 4.1

#### C.2.1  Model Selection

For the pre-trained generative model, we choose the Stable Diffusion XL-Turbo (SDXL-Turbo) model[3] for its fast generation speed and high quality image generation. This model takes text prompts as input and outputs images at a resolution of $512 \times 512$. During our experiments, we use ResNet-50 as the CLIP image encoder backbone. For classifier tuning [18], different text prompt initializations may cause slight differences in accuracy, but since our method focuses on the dataset quality, we simply use the vanilla template "a photo of {class}" for all the datasets.

---

[3] https://huggingface.co/stabilityai/sdxl-turbo

### C.2.2 TRAINING SETTING

Since Stable Diffusion XL-Turbo doesn't use Classifier-free guidance, we simply set the guidance scale to 0 and we set inference steps to 2. For inner training of classification model, we generated 80 images for each class and trained for 20 epochs using the Adam optimizer with a learning rate from $2e-3$ to $2e-5$, equipped with the Cosine learning rate schedule. For outer training, we set the hyper-parameters of the GA algorithm as follows: popsize of 80, maxiter of 80.

Regarding the selection of few-shot datasets, we randomly selected 10 images per class to form the few-shot datasets. For CIFAR10, STL-10, Imagenette, EuroSAT we learn 20 prompts for each class, while for others, we use the technique mentioned in the Section 3.3 and learned 10 common prompts and 10 class-specific prompts for each class. We do training on 8 NVIDIA A800 GPUs, with pytorch 1.12.1 and Ubuntu 20.04.

### C.2.3 EVALUATION SETTINGS

Stable Diffusion model settings are the same in Appendix C.2.2. We generated 800 images for each class and fine-tune CLIP for 30 epochs. We use the the Adam optimizer equipped with the Cosine schedule. After training, we use the fine-tuned CLIP model to do evaluation on real test datasets. All the results are the average over 5 times run, with random seed in 7, 21, 42, 84, 105.

## C.3 EXPERIMENT SETTINGS IN SECTION 4.2

**FakeIt:** FakeIt use Stable Diffusion V1-4 model and different classifier-free guidance scale, but our generative model are not fit for using classifier-free guidance, so we re-implemented their generation approach under our generative model. Other training settings are the same with original paper, including classification model architecture, training learning rate, data augment strategy and so on.

**SuS-X:** The generative model of SuS-X is Stable Diffusion V1-4. For a better performance comparion, we reimplement SuS-X method with SDXL-Turbo model for higher quality image generation. The prompt strategy and other experimental settings keep the setting in the original paper.

**CaFo:** Since CaFo utilizes the OpenAI model to generate description for CLIP text initialization, and the original model has been deprecated, we employed the simple template "a photo of {class}" for text initialization across all datasets to ensure fairness. All other experimental settings remain consistent with the original paper. We have to point that CaFo is a few-shot learning method, and we only report the 16-shot result in Table 2 due to space limitation. Other shot results are given in Section D.5.

# D MORE EXPERIMENTAL ANALYSIS

## D.1 WHY IS ADAPTIVE PROMPT LEARNING NECESSARY?

To validate the necessity of adaptive prompt selection, we implement two prompt selection baseline strategies: (1) randomly selecting the same number of prompts from the prompt pool. (2) Using all prompts of the prompt pool. Table 4 shows the performance comparison on the STL-10 dataset. All the experiment settings are the same as Appendix C.2.3. We can see that the adaptive prompts selected by DeCap method could significantly improve classification model performance compared to random selection strategy. Besides, although using all prompts in the prompt pool offers more sufficient diversity than subset selection, it suffers from various issues mentioned in Section 4.3, which may deteriorates the performance of classification models. This explains that the performance of all prompts is only better than the random selection strategy but not as good as DeCap method. These results further support that adaptive prompt learning strategy is more effective in generating high-quality images for downstream few-shot learning tasks.

Table 4: Comparion of random selection, all selection strategy and DeCap method.

| Random | All | DeCap |
|--------|-----|-------|
| 94.74 | 95.19 | **95.90** |

Table 5: Ablation study on the selecting prompt numbers per class.

| 5 | 10 | 20 | 40 |
|---|----|----|----|
| 95.73 | 95.82 | **95.90** | 95.81 |

Table 6: Ablation study on the number of GA algorithm iterations.

| 20it | 40it | 60it | 80it |
|------|------|------|------|
| 95.73 | 95.87 | **95.91** | 95.90 |

## D.2 ABLATION STUDY

We conducted ablation experiments on two important parameters of our method: the number of prompts selected per class and the iteration count of the GA algorithm. By Table 5, We find that fewer prompts may lead to low dataset diversity, negatively impacting model performance, while more prompts increase optimization difficulty, making it hard to find the optimal solution. We suggest to set the number of prompts selected per class as 20.

Table 6 shows the performance of different numbers of GA algorithm iterations. We observed that performance of classification model converges around 80 generations. In our all experiments, we suggest to set the number of GA algorithm iterations as 80.

## D.3 DIFFERENT METRICS

In this section, we give results of other metrics including precision (Table 7), recall (Table 8) and F1-score (Table 9), which are commonly used in few-shot learning, to further explore the robustness and generalization ability of DeCap. Some brief introduction about these metrics are given as follows:

- **Precision:** Precision measures the accuracy of positive predictions. It is defined as:

$$\text{Precision} = \frac{\text{True Positives (TP)}}{\text{True Positives (TP)} + \text{False Positives (FP)}}$$

Precision answers the question: *Of all the instances predicted as positive, how many are actually positive?*

- **Recall:** Recall, also known as sensitivity or true positive rate, measures the ability of the model to correctly identify positive instances. It is defined as:

$$\text{Recall} = \frac{\text{True Positives (TP)}}{\text{True Positives (TP)} + \text{False Negatives (FN)}}$$

Recall answers the question: *Of all the actual positive instances, how many were correctly predicted?*

- **F1 Score:** The F1-score is the harmonic mean of precision and recall, providing a single metric that balances both. It is defined as:

$$F1 = 2 \cdot \frac{\text{Precision} \cdot \text{Recall}}{\text{Precision} + \text{Recall}}$$

The F1-score is particularly useful when the class distribution is uneven or when precision and recall are equally important.

The results demonstrate that our method performs well on these metrics, indicating that it not only achieves high accuracy but also excels in identifying positive samples and is more cautious when dealing with them. It more comprehensively illustrates the robustness and generalization of our method.

Table 7: **Precision** results of different methods among all datasets.

|  | vanilla | multi | LE | CiP | DeCap |
|---|---|---|---|---|---|
| STL10 | 95.35 | 94.90 | 94.17 | 95.16 | **95.63** |
| CIFAR10 | 76.61 | 76.55 | 77.32 | 77.23 | **77.40** |
| Im-10 | 97.27 | 97.30 | 97.27 | 97.30 | **97.34** |
| Pets | 82.58 | 84.76 | 84.17 | 84.52 | **85.70** |
| Caltech-101 | 84.42 | 84.57 | 85.27 | 85.39 | **85.43** |
| Im-100 | 69.82 | 71.27 | 69.28 | 73.03 | **71.90** |
| EuroSAT | 43.01 | 38.45 | **50.50** | 47.32 | 49.36 |
| Aircraft | 18.38 | 18.85 | **20.85** | 18.67 | **20.85** |
| Country211 | 17.20 | 17.26 | **18.10** | 17.12 | 17.77 |

Table 8: **Recall** results of different methods among all datasets.

|  | vanilla | multi | LE | CiP | DeCap |
|---|---|---|---|---|---|
| STL10 | 95.31 | 94.78 | 94.04 | 94.72 | **95.57** |
| CIFAR10 | 72.39 | 69.63 | 68.24 | 68.63 | **76.99** |
| Imagenette | 97.25 | 97.28 | 97.25 | 97.28 | **97.33** |
| Pets | 81.69 | 82.05 | 82.13 | 83.12 | **84.52** |
| Caltech-101 | 85.21 | 86.02 | 85.37 | 86.07 | 8**6.54** |
| Imagenet100 | 68.32 | 69.98 | 67.00 | 69.56 | **70.94** |
| EuroSAT | 31.07 | 29.62 | 28.31 | 40.62 | **42.22** |
| Aircraft | 17.03 | 17.84 | 17.72 | 17.98 | **19.71** |
| Country211 | 13.72 | 13.90 | 14.14 | 14.98 | **15.44** |

Table 9: **F1-score** results of different methods among all datasets.

|  | vanilla | multi | LE | CiP | DeCap |
|---|---|---|---|---|---|
| STL10 | 95.29 | 94.71 | 93.99 | 94.74 | **95.58** |
| CIFAR10 | 71.89 | 69.04 | 67.30 | 69.19 | **76.89** |
| Imagenette | 97.23 | 97.26 | 97.23 | 97.26 | **97.31** |
| Pets | 81.28 | 81.96 | 82.09 | 83.23 | **84.67** |
| Caltech-101 | 82.04 | 82.87 | 82.12 | 83.45 | **83.73** |
| Imagenet100 | 67.06 | 68.99 | 65.52 | 69.42 | **70.25** |
| EuroSAT | 26.36 | 24.43 | 24.38 | 36.36 | **39.34** |
| Aircraft | 15.10 | 15.98 | 16.00 | 15.95 | **17.81** |
| Country211 | 13.18 | 13.35 | 13.62 | 14.36 | **14.93** |

## D.4 ADVERSARIAL ROBUSTNESS

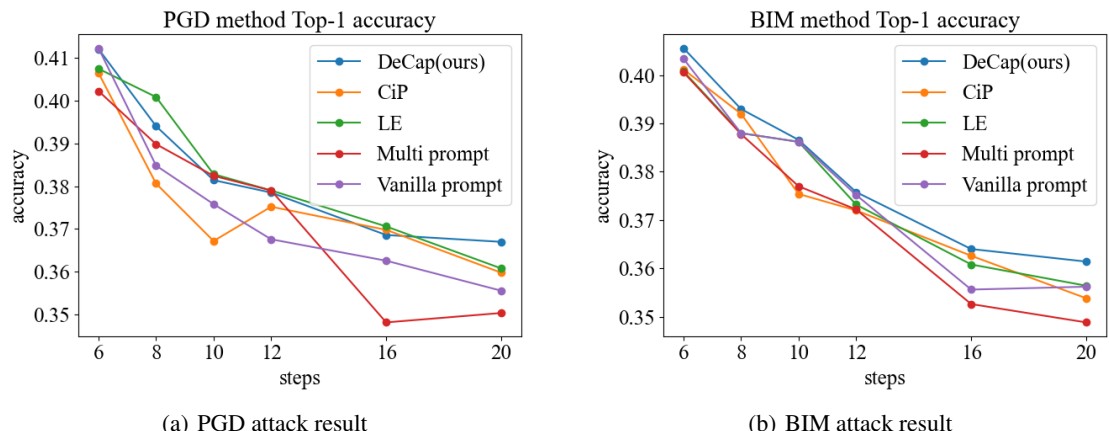

(a) PGD attack result      (b) BIM attack result

Figure 4: **Adersarial robustness of classification models trained with generated images using different prompts designing methods.** We report the results on the ImageNet100 validation set under two adversarial attack methods. The horizontal axis represents the number of steps taken in the attack, and the vertical axis represents the accuracy of the trained classification model on the validation set after the attack.

Adversarial learning aims to evaluate model robustness by adding small perturbations to the input data, causing the model to make false predictions but making little difference to human observers. We use two common attack methods: BIM (Basic Iterative Method) attack [35] and PGD (Projected Gradient Descent) attack [42]. The BIM employs an iterative gradient ascent approach, where at each step, BIM perturbs the image along the gradient direction predicted by the model. It can be written as $x_{i+1} = x_i + \epsilon \nabla_{x_i} J_\theta(x_i, y)$, where $x_0$ denotes the original image, $y$ denotes its label, and $\nabla_{x_i} J$ means the gradient of loss function w.r.t. $x_i$. PGD further projects the adversarial examples into an $\epsilon$-ball around the original image.

We use classification model weights obtained from Section 4.1 and implement adversarial attack on ImageNet100 validation dataset. We use torchattacks [32] library to conduct this experiment. We select attack step size $\epsilon$ as $1/255$ for these two methods, and Fig.4 reports the attack result on different attack steps. We found that model-generated prompts, due to their rich content details, have a slight advantage in adversarial robustness compared to hand-crafted prompts. Moreover, since DeCap integrates the strengths of hand-crafted and model-generated prompts methods, it consistently performs well in terms of resilience against adversarial attacks.

## D.5 MORE EXPERIMENTAL RESULTS OF CAFO AND CAFO + DECAP METHODS

Fig.5 show more experimental results on different shots of each class for CaFo and CaFo + DeCap Methods. The experimental results are aligned with conclusions in Section 4.2.

## D.6 ARE PROMPTS WITH DOMAIN INFORMATION ENOUGH FOR CLASSIFICATION?

To illustrate this point, we conducted experiments on the Sketch subclass of the PACS dataset [39]. In this dataset, all images follow the same style. As shown in Fig.6, the hand-crafted prompts could generate sketch-style guitar images, while the image distribution deviates the distribution of real images. This could explain the degraded performance of hand-crafted prompts methods. This is aligned with existing substantial theory

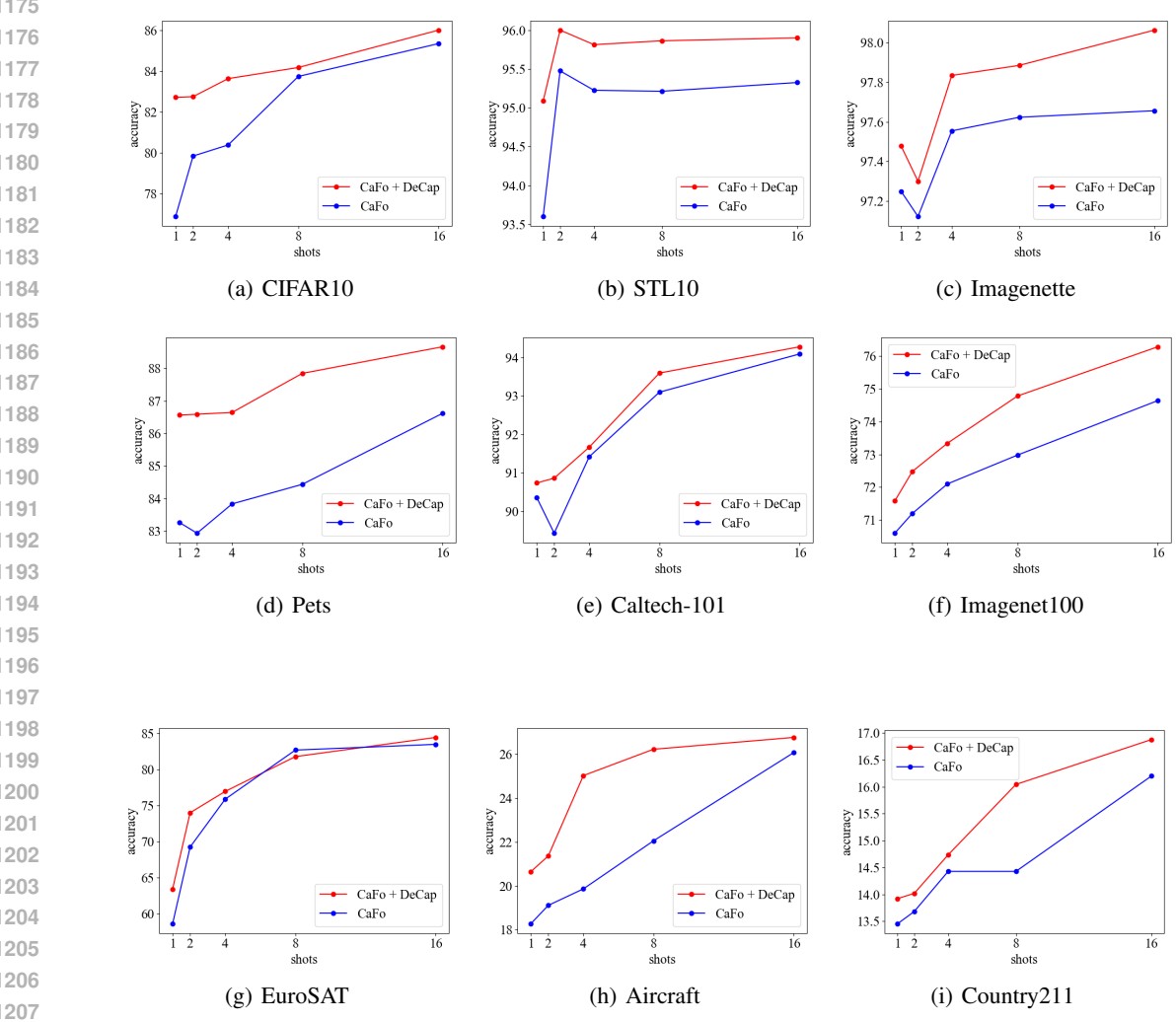

Figure 5: Classification accuracies on different shots of each class for CaFo and CaFo + DeCap Methods

[15; 83; 104], which suggests that samples perfectly matching the real data distribution are most useful for classification. As a comparison, the synthetic images by DeCap method are surprisingly composed of only a portion of sketch-type prompts, supplemented by a significant amount of other types of prompts. The discrepancy between synthetic images and real images is significantly large, however, the performance of classification model training with synthetic data using DeCap method could approach the performance with real data. This result cannot be well explained by existing theories. We hope that a rational theoretical insight could characterize such phenomenon in the future study.

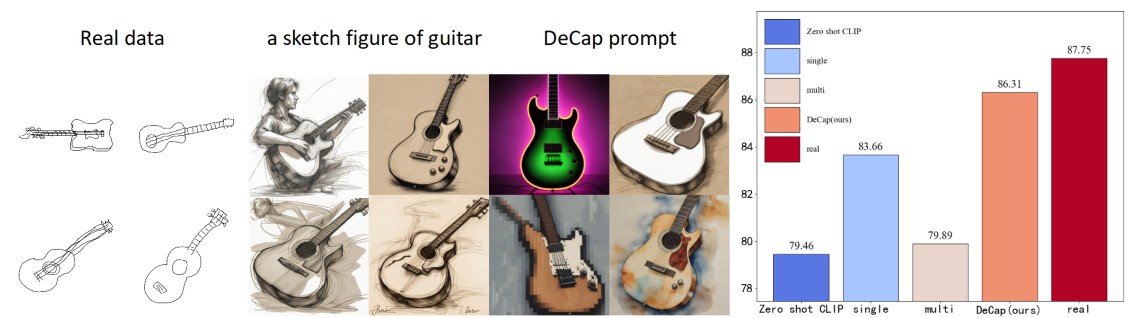

Figure 6: (Left) Examples of real images, synthetic images by hand-crafted prompts and DeCap methods on PACS Sketch dataset. (Right) Performance comparison between hand-crafted prompts and DeCap methods.

# E    VISUALIZATION OF SYNTHETIC IMAGES AND LEARNED PROMPTS

## E.1    VISUALIZATION OF SYNTHETIC IMAGES

Fig.7 and Fig.8 shows some examples of synthetic images on Pets and Imagenet100 datasets by DeCap method.

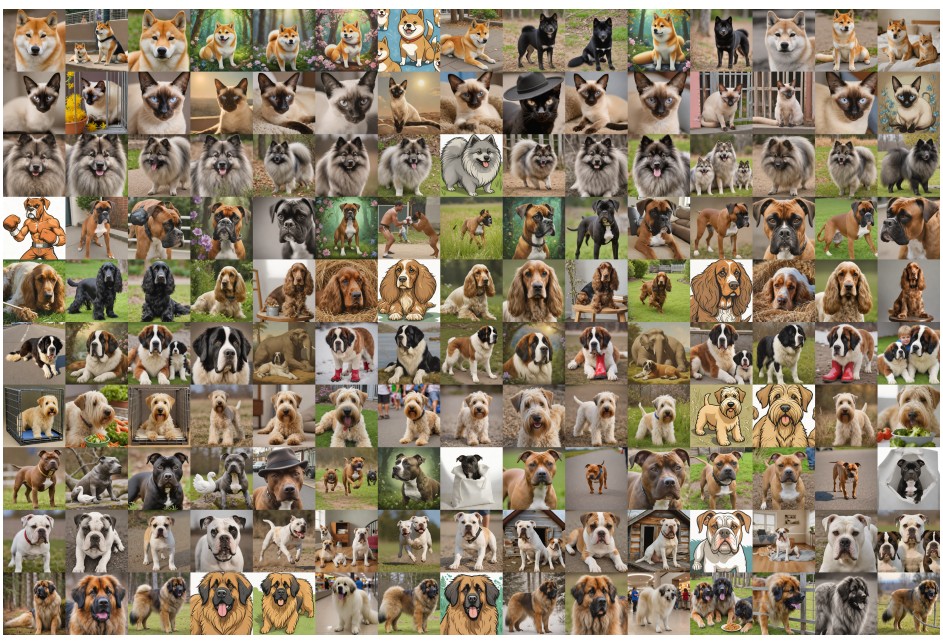

Figure 7: Examples of generated images on Pets dataset by DeCap method.

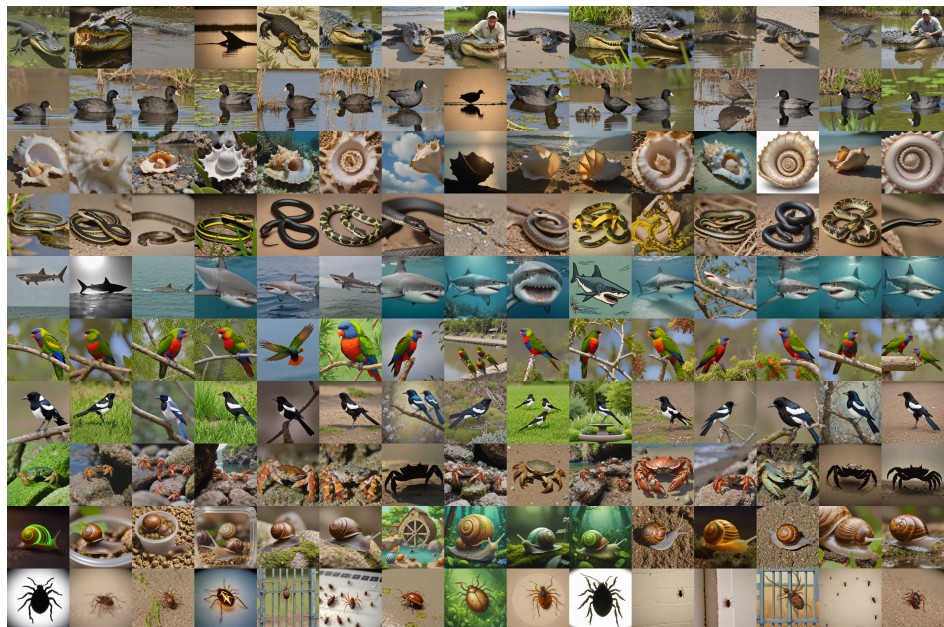

Figure 8: Examples of generated images on Imagenet100 dataset by DeCap method.

## E.2 VISUALIZATION OF LEARNED PROMPTS

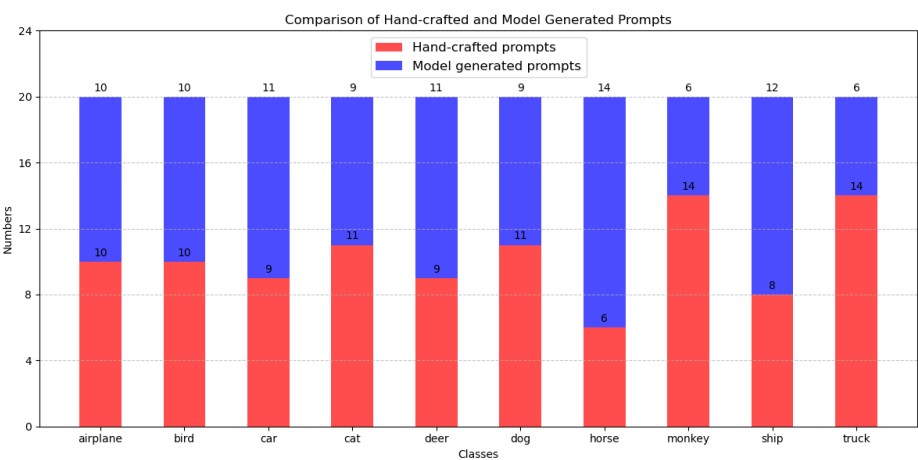

Figure 9: Illustrations of the number of hand-crafted prompts vs the number of model-generated prompts mined by DeCap method on STL-10 dataset.

We will demonstrate that DeCap method can adaptively learn proper and dataset-specific prompts that are suitable for concerned tasks from the following three aspects.

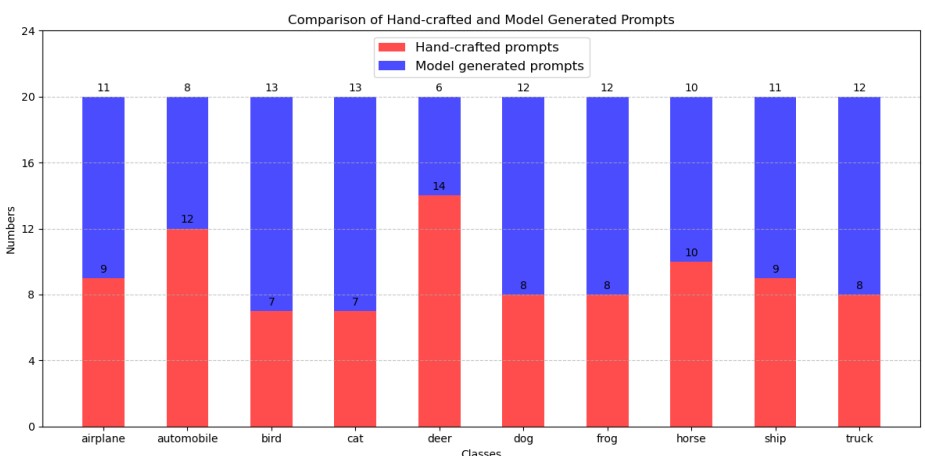

Figure 10: Illustrations of the number of hand-crafted prompts vs the number of model-generated prompts mined by DeCap method on CIFAR10 dataset.

Firstly, the ratios of the number of model-generated and hand-crafted prompts for each class are varying, as shown in Fig. 9 and 10. This reflects that our method could adaptively adjust the proportions that reconcile class/domain information and rich content information for different classes.

Secondly, though the hand-crafted prompts follow the same templates, we can see that different classes may learn relatively different prompts in Fig.11. This further reveals our method could adaptively learn classification-aware prompts for each class, so as to achieve better performance on downstream tasks. Moreover, we additionally give some examples about the consistently selected model-generated prompts during the optimization process to further highlight the significance of integrating fine-grained prompt descriptions. As we can see in Table 11, the consistently selected prompts show high diversity and fine-grained information, including: movement, posture, background, color, quantity, other objects, and so on. This pictures significantly help to provide classification-benefit features.

Lastly, Table 10 shows that though STL-10 and CIFAR-10 datasets have some same categories, the learned prompts by our method could be almostly different. This demonstrated that our DeCap method could learning proper prompts suitable to concerned few-shot datasets. For instance, we can see that learned prompts for the STL-10 dataset are realistic, while learned prompts for CIFAR-10 dataset are of low-resolution imagery. Notice that these prompts are well aligned with prior knowledge of these datasets.

Moreover, we additionally display the complete set of prompt pool of the "airplane" class in STL-10 dataset in Table 12, to offer a more intuitive understanding for the characteristic of our method stated above. And we further give visualizations that demonstrate the prompt selection process over the course of optimization, including image examples and the evolution of prompts, please see Fig.12.

Table 10: Illustration of mined prompts for "deer" class on different datasets. DeCap method selects completely different prompts for the same class across different datasets, demonstrating its ability to adaptively learn the prompts suited to each specific dataset.

| |
|---|
| A deer is grazing the woods. |
| deer are grazing under a tree |

29

STL-10

| | |
|---|---|
| | a photo of the clean deer. |
| | A silhouette of deer. |
| | a deer and young man roam around during a december game |
| | a photo of a deer. |
| | a deer in a video game. |
| | a toy deer |
| | a deer on a pond |
| | the cartoon deer. |
| | the hornets and deer are on a ridge |
| | a deer. |
| | deer resting with the grazing padou atop old farmhouse |
| | An ink painting of a deer |
| | deer on a green pond. |
| | the toy deer. |
| | a brown bear eats the deer |
| | A glossy deer. |
| | a photo of a large deer. |
| | a group of deer on prairie are seen grazing in their natural habitat |
| | a photo of deer, a wild deer in the wild |
| | a deer on a farm |
| | A soft-focus deer. |
| | art of a deer. |
| | deer and their prey on the northern slopes |
| | a photo of deer, a deer standing in the snow with a sky background |
| | a pixelated photo of the deer. |
| | several deer grazing in the desert |
| | fox and a deer on the grounds of a city |
| CIFAR10 | a rendering of a deer. |
| | a photo of deer, a group of deers standing in a field |
| | A silhouette of deer. |
| | a photo of deer, a deer is standing in the grass |
| | a deer is grazing an ancient inscription. |
| | a photo of deer, a herd of deer in the desert |
| | deer and the munro. |
| | A pair of deer on a trail. |
| | a hunt deer on a desert land |
| | deer and the munro. |
| | a pixelated photo of the deer. |

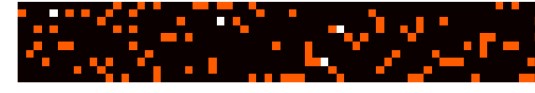

(a) CIFAR-10 dataset        (b) STL-10 dataset

Figure 11: Illustration of mined hand-crafted prompts for each class by DeCap method across two datasets. Each column represents the same template, and each row indicates which prompts were selected for class of this row. Black indicates prompt is not selected, orange indicates the prompt is selected once, and white indicates the prompt is selected more than once. For clarity, we removed prompts that were not selected by any class of the dataset. It could be observed that although the prompt templates are the same, the domain information required by each class is distinctly different, demonstrating DeCap's ability to adaptively learn suitable prompts for the classification of each class.

Table 11: Examples of model-generated prompts which are consistently selected during optimization process.

| a bird sitting on a branch. | cars that have to make an effort to turn off. | A black cat is in a room where the window is down. |
|---|---|---|
| 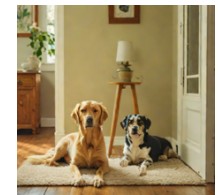 | | |
| A truck with lots of people on it. | A deer is grazing the woods. | A dog is standing in its yard with a harness on it. |
| | | |
| A white horse in the open barn. | dogs inside a home on a summer. | a semi truck driving down a rural road. |
| | 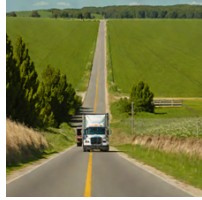 | |

Table 12: The prompt pool of "airplane" class in STL-10 dataset. Mined prompts by DeCap method are highlighted in **bold**.

| Model-generated prompts |
|---|

| | | | |
|---|---|---|---|
| a photo of airplane , a small plane is parked on the runway | a photo of airplane , a large passenger jet flying through a blue sky | a photo of airplane , a small plane is on the runway | a photo of airplane , a yellow airplane flying through a blue sky |
| a photo of airplane , a large passenger jet flying through the sky | **a photo of airplane , a small plane flying in the sky** | a photo of airplane , a small plane is floating in the water | a photo of airplane , a large passenger jet sitting on a runway |
| a photo of airplane , a small plane flying in the sky | a photo of airplane , a small blue airplane is taking off from the runway | a photo of airplane , a plane is parked on the tarmacl | a photo of airplane , two small planes are sitting on the water |
| **a photo of airplane , a plane flying in the sky** | a photo of airplane , a small plane flying over a mountain range | a photo of airplane , a plane is on the runway | a photo of airplane , a small plane flying through the air |
| a photo of airplane , a large white plane | a photo of airplane , two planes flying in the sky | a photo of airplane , a plane flying in the sky | a photo of airplane , a small plane flying over a city |
| a photo of airplane , a plane flying in the sky | a photo of airplane , a small plane flying through the air | a photo of airplane , a plane flying in the sky | a photo of airplane , a plane is parked on the tarmacl |
| a photo of airplane , a plane flying in the sky | a photo of airplane , a plane flying in the sky | a photo of airplane , a small plane is parked on the water | a photo of airplane , a plane flying in the sky |
| a photo of airplane , a small plane sitting on a snowy field | a photo of airplane , a small plane flying in the sky | An airplane that has been seen flying over another airplane. | A plane is in a parking lot. |
| airplane that you bought a few years ago | A small airplane is flying over a highway at a time. | **An airplane that is parked in an airport** | The plane has an engine, a seat, a console, a charger, and |
| An aircraft is in the flight over a lake. | The airplanes are all parked inside the parking lot. | A plane is in the air. | plane of a small aircraft. |
| A red and white airplane with a green and green color scheme. | An airplane parked on the runway near a pier. | An airplane that has just broken ground behind it. | A plane parked next to one of the airplanes above it's engine. |
| **Airplanes in space that are not as big as usual.** | An airplane parked along a highway. | **A small airplane that's flying at low speeds under a cloudy sky.** | airplanes need people to work hard at the zoo |
| **An airplane parked next to a bridge.** | Some airplanes flying over people. | The airplane is in a green sky with blue skies. | The airplane with the lights is about to be docked. |
| An airplane with three engines and a propeller. | **an airplane with a window** | An airplane parked on top of a hill next to it | airplane on the tracks. |
| **An airplane on an airplane track** | A plane with tires on it flying away from it. | An airplane is parked on a runway at a airport. | These airplanes are in a wing. |
| a commercial airplane traveling in july. | airplane in flight... a photo and video | Two air planes all flying in a row. | A modern airplane is arriving in the air. |
| An airplane in the middle of nowhere with its doors lowered. | airplane is parked in a parking lot | An airplane that is coming in to land. | passengers in an airplane in the rain |

| airplanes flying at a rate of 2 to 3 mph on a sunday | An airplane on a runway next to a small green field. | an airplane on an airport runway | a classic red blue airplane is shown in the cockpit with bright colors as well. |
|---|---|---|---|
| planes in a dry pit | airplane and other objects in the air | an airplane makes an outgoing landing on the ground | An old airplane is coming down the track. |
| A man attempting to board a commercial airplane. | Small airplanes with wing lights attached to them. | A small airplane with the tail mounted up. | An airplane in a flight path with some passengers nearby. |
| airplane on an old building | An aircraft goes up through a window dripping with smoke and debris. | airplanes that have been converted to jet engines | airplanes cruising in the bay. |
| A boy is running with an airplane that is on the runway. | A blue airplane has its wings shut. | An airplane is about to land in a parking lot and be delivered. | two airplanes parked at the airport |
| A white airplane on the runway with blue ice. | jet airplane is ready for a test | An airplane is sitting on a ground with all three engines on the ground. | There's one airplane in the cockpit which is parked by another airplane. |
| An airplane is in the air. | A family is on a small airplane at a hotel. | aircraft carrier and an airplane together with some gulls. | airplane inside of the airplane |
| The airplane is looking down. | An airplane is shown flying on a runway. | small bodied airplane on a plane | An airplane parked next to fireworks on the sky. |
| An airplane that appears to be on the runway. | An airplane that is very close to the ground in an airport. | Various aircraft and airplanes are getting ready for flight. | an airplane is seen arriving on a runway |
| Two aircrafts in a white airplane at a station. | The airplane landed. | an airplane that is making a flying flight | airplane on the runway at the airport |
| this airplane was able to take off with just a small amount of effort to get the | An airplane that is in a flying position. | An airplane making its way between jets. | airplane sitting in air |
| a large old plane sits off the fuel tank | aircraft carrier and its crew arriving in an airplane | airplane on the runway | A family of airplanes are in a building. |
| plane flies around city | an airplane about to land in a desert | An electric airplane in the sky. | **an airplane that is making it's way around the tarmac** |
| airplanes on the runway | **the crew of airplane on board** | The airplane is in the air. | jet airplane wing during maintenance |
| A blue and white airplane with white wing panels. | A commercial airplane flying under the radar. | The airplane has been damaged by the winds. | An airplane flying near a tarmac. |
| Hand-crafted prompts | | | |

| | | | |
|---|---|---|---|
| a good photo of the airplane. | a photo of many airplane. | a sculpture of a airplane. | a photo of the hard to see airplane. |
| a low resolution photo of the airplane. | a rendering of a airplane. | graffiti of a airplane. | a bad photo of the airplane. |
| a cropped photo of the airplane. | a tattoo of a airplane. | the embroidered airplane. | a photo of a hard to see airplane. |
| a bright photo of a airplane. | a photo of a clean airplane. | a photo of a dirty airplane. | **a dark photo of the airplane.** |
| a drawing of a airplane. | a photo of my airplane. | the plastic airplane. | a photo of the cool airplane. |
| a close-up photo of a airplane. | a black and white photo of the airplane. | a painting of the airplane. | a painting of a airplane. |
| a pixelated photo of the airplane. | **a sculpture of the airplane.** | a bright photo of the airplane. | a cropped photo of a airplane. |
| a plastic airplane. | **a photo of the dirty airplane.** | a jpeg corrupted photo of a airplane. | a blurry photo of the airplane. |
| a photo of the airplane. | a bad photo of a airplane. | a rendering of the airplane. | a airplane in a video game. |
| a photo of one airplane. | a doodle of a airplane. | a close-up photo of the airplane. | a photo of a airplane. |
| the origami airplane. | **the airplane in a video game.** | a sketch of a airplane. | **a doodle of the airplane.** |
| a airplane. | a origami airplane. | a low resolution photo of a airplane. | the toy airplane. |
| a rendition of the airplane. | a photo of the clean airplane. | a photo of a large airplane. | **a rendition of a airplane.** |
| a photo of a nice airplane. | a photo of a weird airplane. | a blurry photo of a airplane. | a cartoon airplane. |
| art of a airplane. | a sketch of the airplane. | a embroidered airplane. | a pixelated photo of a airplane. |
| a jpeg corrupted photo of the airplane. | a good photo of a airplane. | a photo of the nice airplane. | **a photo of the small airplane.** |
| a photo of the weird airplane. | the cartoon airplane. | art of the airplane. | a drawing of the airplane. |
| a photo of the large airplane. | a black and white photo of a airplane. | a dark photo of a airplane. | graffiti of the airplane. |
| a toy airplane. | a photo of a cool airplane. | a photo of a small airplane. | a tattoo of the airplane. |
| a digital style airplane | a colorful airplane | a modern style airplane | an abstract photo of airplane |
| a cartoon style airplane | a virtual style airplane | An ink painting of a airplane | a toy airplane |
| **A model airplane.** | a red airplane | a blue airplane | a yellow airplane |
| a black airplane | a white airplane | An old airplane. | A futuristic airplane. |
| A minimalist airplane. | A detailed illustration of airplane. | A close-up of airplane. | A shadowy figure of airplane. |
| A silhouette of airplane. | A bright and vibrant airplane. | An abstract concept of airplane. | A vintage style airplane. |

| A neon-lit airplane. | A monochrome airplane. | A watercolor painting of airplane. | A sketch of airplane. |
|---|---|---|---|
| A digital art of airplane. | A handcrafted airplane. | An aerial view of airplane. | A side profile of airplane. |
| A textured airplane. | A glossy airplane. | A matte airplane. | A glowing airplane. |
| **A rustic airplane.** | A weathered airplane. | A sparkling airplane. | A serene airplane. |
| A chaotic airplane. | A whimsical airplane. | **A dynamic airplane.** | A frozen moment of airplane. |
| A soft-focus airplane. | A high-contrast airplane. | A sepia-toned airplane. | A saturated airplane. |
| An isolated airplane. | A mirrored airplane. | A panoramic view of airplane. | An enchanted airplane. |

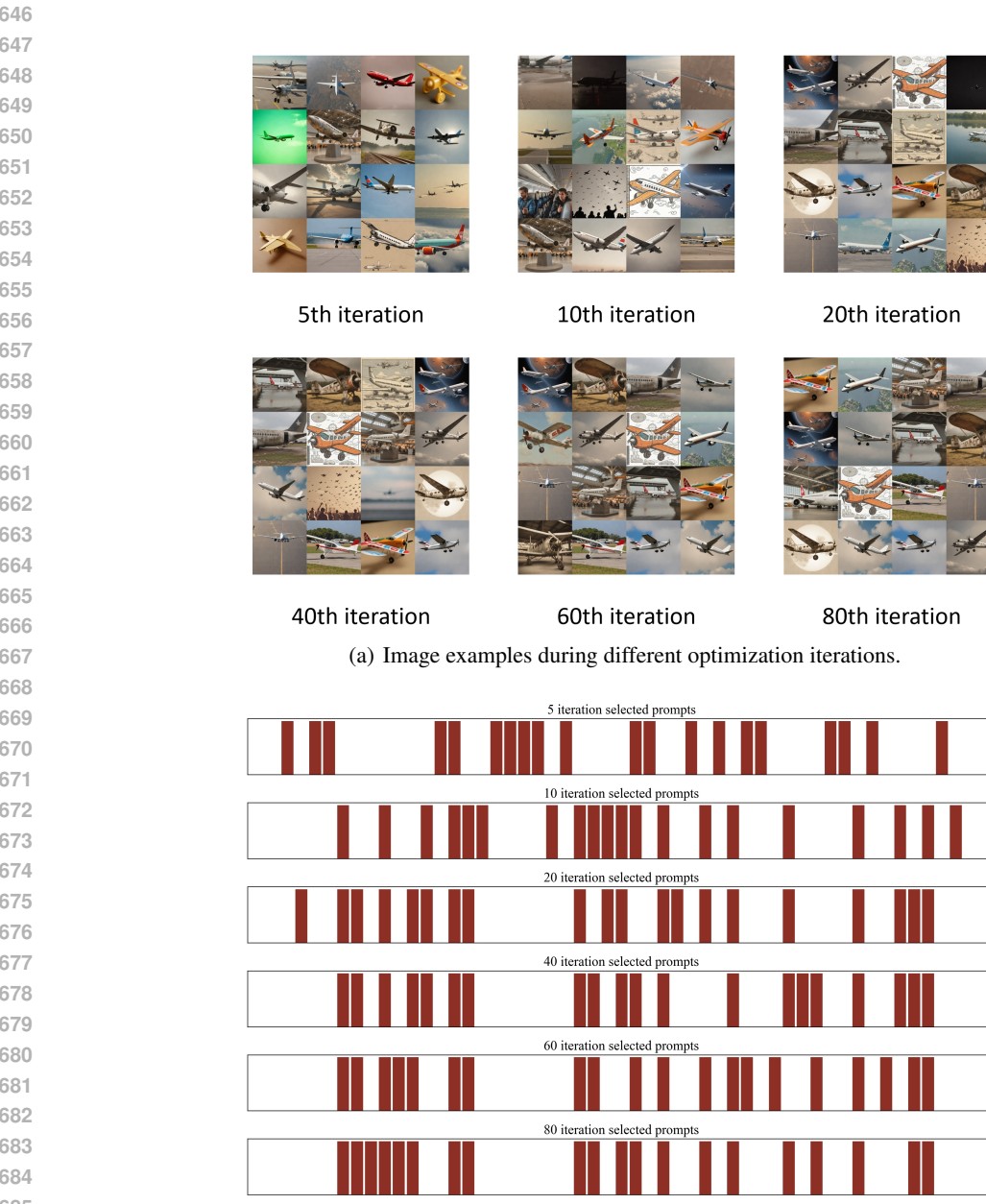

(a) Image examples during different optimization iterations.

(b) The evolution of prompts during different optimization iterations.

Figure 12: Image examples and the evolution of prompts during different optimization iterations. For clarify, Fig.(b) shows only the selected prompts and omits the rest.

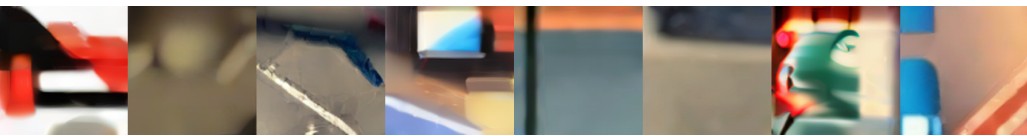

(a) 96 resolution

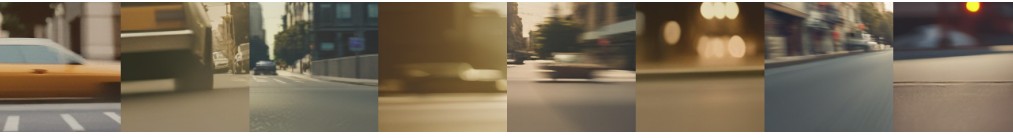

(b) 224 resolution

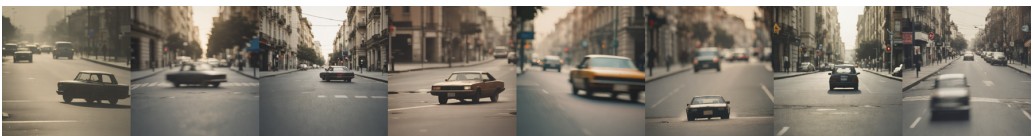

(c) 512 resolution

Figure 13: prompt: "a photo of a car in the street"

## F REBUTTAL DISPLAYS

This section is just for additional rebuttal visualization.

### F.1 DIFFERENT RESOLUTIONS

We give some examples about generating images using different resolution in Fig.13. We can see that generative large models are only good at the resolution of its training set, i.e. 512 resolution.

