# OpenReview forum: "Diversity-Enhanced and Classification-Aware Prompt Learning for Few-Shot Learning via Stable Diffusion"
_ICLR.cc/2025/Conference — Submitted to ICLR 2025_

### Official Review · Reviewer_9gWJ · 2024-10-18

**Soundness:** 1
**Presentation:** 3
**Contribution:** 2
**Rating:** 3
**Confidence:** 5

**Summary:**

Authors tackle the problem of synthetic data generation for downstream training. They look at this from the perspective of increasing prompt diversity, to increase image diversity. This do this by generating proper prompts through meta learning of the prompt pool. They evaluate this against other prompt-learning methods and on top of SOTA methods.

**Strengths:**

- Well written

- Novel methodology

- Good show of qualitative results

**Weaknesses:**

Weaknesses are given in order of how I perceive their importance (issues that weigh heavier in my review decisions are first):

1) I find the main experimental results comparing to SOTA prompt methods in Table 1 lacking (specifically in terms of datasets), for the following reasons:

1a) The number of datasets evaluated is lacking very far behind other SOTA methods. E.g. Sus-X (cited in this paper) trains on 19 datasets, only three of which are used here (and two subsets). Furthermore, the datasets used in DeCap are quite similar in terms of subject, granularity, modality, etc. This is concerning because the problem statement of generating training images is highly class / dataset dependent. To be rigorously shown, this needs to be evaluated on a diverse variety of datasets. I believe that to support the claims in this paper, this method would need to be evaluated on a minimum of 3 / 4 of the following categories: a FG dataset (e.g. Stanford Cars of FGVC Aircraft), two more 'types' of objects (e.g. Food101, FGVC Aircraft, assuming animals has already been covered by Pets), a different modality (e.g. EuroSAT, ImageNet-Sketch), a scene-based dataset (e.g. SUN397). I would like to reiterate that I believe this is the minimum, and more ideally a few more strategically chosen datasets should be evaluated.

1b) CIFAR and STL are 32x32 and 96x96 resolutions, respectively. SDXL-Turbo generates images at 512x512. Hence, they are too far out-of-distribution for how the diffusion model generates to be strong evidence in themselves. I don't find it problematic in itself to include them, but these datasets being weak further exacerbates the issue described in 1a.

2) Intuitively, DeCap looks expensive. I'm not convinced that the performance gains are enough to justify the cost (unless you correct me on the cost). This would be stronger if the GPU hours were compared between DeCap and other SOTA in table 1 (LE, CiP) in addition to how much it adds for other SOTA methods (FakeIt, SuS-X, CaFo).

3) When combining DeCap with SOTA, significant gains are only seen on FakeIt. Already SuS-X and CaFo (more recent works) show gains of only 1% or less on average. There are two outcomes that I am concerned these results imply at least one of the following:

3a) FakeIt is trained from scratch, while the other two fine-tune SOTA models. It could be that in the context of fine-tuning pre-trained models, DeCap does not affect as much.

3b) FakeIt is older and both SuS-X and CaFo are newer. It could be that the challenges DeCap combats have been already well-addressed by newer works.

4) The work misses ablations on the design choices. E.g. why not optimize the prompt without meta learning, choice of loss function, choice of meta learning algorithm, etc.

5) The ablations on number of prompts, etc. are done on STL-10 (see 1b)

**Questions:**

1) What is your justification for dataset selection? Does it align with previous work?

2) What is the cost of DeCap? I'm not finding the breakdown, but please point me to it if it is already there.

---

> ### Author Response · Authors · 2024-11-24
> **Response Part1/3**
>
> # More datasets should be inculded
> ## Response 4.1:
> As suggested by the reviewer, we have re-executed experiments on the EuroSAT and FGVC Aircraft datasets, which cover OOD, fine-grained, and object classification. We are also working on the Country211 scene dataset. Please allow us some time to complete this. The results are presented in Table 1,2 of Section 4.1 of the manuscript. We also list them here:
> | Dataset   | Vanilla | Multi  | LE    | CiP   | DeCap |
> |-----------|---------|--------|-------|-------|-------|
> | EuroSAT   | 31.31   | 30.11  | 29.35 | 39.91 | 41.94 |
> | Aircraft  | 17.04   | 17.85  | 17.73 | 18.00 | 19.74 |
>
> As it shown, proposed DeCap method could consistently imporve baselines across these datasets. These experimental results further support the capability of proposed DeCap method in generating high-quality images for downstream few-shot learning tasks.
>
> # The OOD issues caused by resolution.
> ## Response 4.2:
> We want to kindly clarify that the image resolution is not a major issue.
> In practive, we could resize the images generated by the model to 32 or other resolutions without causing significant OOD issues compared to the original datasets. Additionally, the fact that our method still works well on low-resolution datasets further emphasizes the effectiveness of our approach for different types of datasets.
>
> Moreover, the difference between a 96 resolution image and the standard 224 resolution image do not result in significant OOD effects. To illustrate this, we conducted an experiment where we resized the data from the original resolution to 96, then further resized it to 224 on the ImageNet100 dataset. The results showed that the performance only changed by about 0.1\%, which we believe is entirely acceptable.

---

> > ### Author Response · Authors · 2024-11-24
> > **Rsponse Part 2/3**
> >
> > # The cost of DeCap.
> > ## Response 4.3:
> > Firstly, we want to kindly clarify that our method is relatively time-consuming during the meta-training phase, as discussed in Appendix A (limitations), while is high efficiency and generalization during the meta-test stage.
> >
> > 1.We are glad that you acknowledge the innovation of our approach. We would like to reframe our method to provide a clearer understanding of its generalizability and cost. Existing approaches that approximate the data domain are still limited to approximations at the data level, which, in few-shot settings, often lead to overfitting. This results in generating images that are too similar to the few-shot data. Therefore, we needed to rethink the way few-shot data is utilized, moving beyond just visual similarity and focusing on the task-level utilization of this data to improve classification performance. As validated in Fig.6 of Appendix D.6, the prompts we learned lead to images with significantly different visual features from the real domains, but when used in classification, the synthetic data's accuracy approaches that of models trained on the full dataset. Furthermore, we compare our method with domain-approximating methods, and the experiments clearly show the superiority of task-level data generation.
> >
> > | Methods     | SDEidt | Textual inversion | DeCap |
> > |-------------|--------|:-------------------:|-------|
> > | STL10       | 94.74  | 94.86             | 95.9  |
> > | Caltech-101 | 85.82  | 84.76             | 88.67 |
> >
> > To achieve this goal, meta-learning naturally fits into task-level guidance for generating classification-aware datasets. We achieve an adaptive sample selection process that links generated datasets with downstream learning, which is challenging for domain-approximation methods. In meta-learning, we emphasize generalization and efficiency during the meta-test phase. In our case, generalization is demonstrated by the ability to scale from accessible few-shot datasets to full datasets, and efficiency is reflected in the fact that the learned prompts can be directly applied for generalization without any further training. This means when we finish training, our method can enjoy the same speed as hand-crafted methods. Therefore, during the meta-test phase, our method undoubtedly demonstrates high efficiency and generalization. Of course, we also acknowledge that our method is still relatively time-consuming during the meta-training phase, as discussed in Appendix A (limitations). However, the advantage of task-level data generation is something that existing methods cannot achieve.
> >
> > 2.DeCap spends the majority of its runtime on algorithm solving. However, this is currently an alternative to gradient backpropagation. With further advancements in large-scale generative models and their using cost decreasing, our framework can be easily transformed into a white-box optimization model, which will significantly reduce runtime. Currently, we have to retrain the inner loop each time, but with white-box optimization, we can continue training with the weights from the previous loop. By that time, the time cost will no longer be an issue. Our current algorithm has already proven the feasibility of the approach.
> >
> > 3.We present results from other synthetic datasets on the CaFo and SuS-X methods, which show that our method still outperforms the current SOTA methods. This strongly supports the generalizability of our approach and highlights its potential application value.
> > ## SUS-X
> > |        | stl10 | cifar10 | imagenette | pets  | caltech-101 | imagenet100 | average     |
> > |--------|-------|---------|------------|-------|-------------|-------------|-------------|
> > | vanilla | 95.24 | 72.77   | 98.24      | 79.64 | 84.57       | 69.96       | 83.40 |
> > | multi  | 95.29 | 72.28   | 98.24      | 78.99 | 84.89       | 70.08       | 83.30      |
> > | LE     | 95.39 | 73.62   | 98.37      | 79.2  | 84.51       | 69.98       | 83.51 |
> > | CiP    | 95.21 | 73.19   | 98.34      | 79.77 | 84.76       | 70.14       | 83.57 |
> > | DeCap  | 95.43 | 75.89   | 98.39      | 80.4  | 84.89       | 70.3        | 84.22 |
> > ## CaFo
> > |        | stl10 | cifar10 | imagenette | pets  | caltech-101 | imagenet100 | average     |
> > |--------|-------|---------|------------|-------|-------------|-------------|-------------|
> > | vanilla | 95.33 | 85.34   | 97.66      | 86.62 | 94.09       | 74.64       | 88.95 |
> > | multi  | 95.48 | 85.76   | 97.88      | 86.92 | 94.27       | 75.58       | 89.32      |
> > | LE     | 95.51 | 86      | 97.78      | 88.06 | 94.03       | 75.00          | 89.40 |
> > | CiP    | 95.33 | 84.05   | 97.88      | 87.38 | 94.27       | 75.92       | 89.14 |
> > | decap  | 95.9  | 86      | 98.06      | 88.66 | 94.28       | 76.28       | 89.86 |

---

> > > ### Author Response · Authors · 2024-11-24
> > > **Response Part 3/3**
> > >
> > > # SuS-X and CaFo show gains of only 1\% or less on average. Two concerns: In the context of fine-tuning pre-trained models, DeCap does not affect as much. The challenges DeCap combats have been already well-addressed by newer works.
> > > ## Response 4.4:
> > >
> > > 1.In Response 4.3, we provide results showing the improvement brought by different methods. As seen, compared to existing methods, our approach achieves better performance. This 1\% improvement is not easy to achieve.
> > >
> > > 2.The SuS-X and CaFo methods only use synthetic datasets as supplementary information. For example, SuS-X uses the similarity between the synthetic data and the test data as a small part of its metrics, while CaFo incorporates additional information from models like CLIP and DINO. Because these methods only utilize the generated data in a limited way, replacing the synthetic dataset has little impact on performance.
> > >
> > > 3.We apologize for any misleading statements in our paper that may have caused confusion. In fact, the classifier tuning we use is a state-of-the-art algorithm that also fully leverages the synthetic data, and this approach is not meant to suggest that DeCap performs poorly for all fine-tuning-based methods as mentioned in 3a. Regarding 3b, it’s simply because the algorithmic properties of SuS-X and CaFo that makes synthetic data leading to few significant improvements. This doesn't imply that the challenges have been fully addressed. Since we have found that when fully utilizing synthetic data information with the classifier tuning method, DeCap still provides significant improvements.
> > >
> > > # The work misses ablations on the design choices.
> > > ## Response 4.5:
> > > We may not have fully understood your question, and our explanation might not directly address what you intended to ask. If there are any remaining doubts, please feel free to describe the issue more thoroughly.
> > >
> > > 1.Our goal is to find the most suitable dataset for downstream tasks, and for this purpose, we aim to establish a connection between dataset construction and downstream learning from the task perspective. Rather than using meta-learning directly for prompt learning, traditional methods of prompt learning often only generate images that are very similar to real few-shot samples due to the scarcity of real samples. This limits the data expansion to an approximation at the sample level. Therefore, we propose a meta-learning algorithm to adaptively select prompts, achieving a task-aware classification effect, which traditional prompt learning methods struggle to achieve.
> > >
> > > 2.For the selection of the loss function, we follow the consensus in the field: cross-entropy loss. However, we also provide common few-shot evaluation metrics, such as precision, recall, and F1 score in Table 7,8,9 of Appendix D.3.
> > >
> > > 3.Are you referring to the optimization algorithm? Due to the large parameter size of existing generative models, gradient-based algorithms often fail. We have tried several common black-box optimization algorithms, including SPSA and CMA-ES, but unfortunately, they were unable to optimize successfully.

---

> > > > ### Comment · Reviewer_9gWJ · 2024-11-25
> > > > **Clarification 3 / 3**
> > > >
> > > > 1. I'm just missing why the meta-learning is needed--is there not a simpler way to learn prompts? (e.g. via textual inversion)
> > > > 2 / 3. Fair enough, I am fine with this

---

> > > > > ### Author Response · Authors · 2024-11-29
> > > > > **Response to Clarification 3 / 3**
> > > > >
> > > > > Traditional prompt-learning methods based on images, such as Textual Inversion and Custom Diffusion, focus on approximations few-shot domain at the pixel level. However, as highlighted in Response 4.3, this approach is inefficient in few-shot settings. To better connect with downstream tasks, a simple idea might be to approximate at the feature level instead of the pixel level—for example, by calculating the similarity between synthetic and real data through an image encoder.
> > > > > We explored this approach but found it unsuccessful, as it still essentially overfits the real data. Even if this idea were successful, the way it utilizes real data would still fall short in terms of generalization compared to our task-level modeling approach, as guaranteed by meta-learning theory [1-5]. From this perspective, adopting meta-learning becomes a natural choice (though we also welcome other modeling strategies for improve task-level generalization).
> > > > >
> > > > > Moreover, if we view the inner-loop optimization in our method (Eq. 3) as a sophisticated metric, our method aligns conceptually with the ideas behind Textual Inversion. In this sense, our approach extends the Textual Inversion techniques.
> > > > >
> > > > > Finally, meta-learning never means a "not simple" method. It's just the current limitations that make us fail to optimize efficiently ---- we really struggled for alternative optimization methods, but we failed (including approximate implicit differentiation [6], iterative differentiation [7], SPSA [8], CMA-ES [9] and so on).
> > > > >
> > > > > [1].On the theory of transfer learning: The importance of task diversity. NeurIPS. 2020
> > > > >
> > > > > [2] Transfer learning for nonparametric classification: Minimax rate and adaptive classifier. The Annals of Statistics,2021
> > > > >
> > > > > [3] Transfer learning for high-dimensional linear regression: Prediction, estimation, and minimax optimality. Journal of the Royal Statistical Society, Series B, 2022.
> > > > >
> > > > > [4] Learning an Explicit Hyper-parameter Prediction Function Conditioned on Tasks，JMLR, 2023.
> > > > >
> > > > > [5]Metalearning with very few samples per task. COLT, 2024.
> > > > >
> > > > > [6] Optimizing Millions of Hyperparameters by Implicit Differentiation. PMLR, 2020.
> > > > >
> > > > > [7] CMW-Net: Learning a Class-Aware Sample Weighting Mapping for Robust Deep Learning. TPAMI, 2023.
> > > > >
> > > > > [8] Fine-Tuning Language Models with Just Forward Passes. NeurIPS, 2023.
> > > > >
> > > > > [9] Black-Box Tuning for Language-Model-as-a-Service. ICML, 2022.

---

> > > > > > ### Comment · Reviewer_9gWJ · 2024-12-02
> > > > > > **Final Response**
> > > > > >
> > > > > > Dear Authors,
> > > > > >
> > > > > > I would like to thank you kindly for all your work and efforts, and for your openness in this discussion.
> > > > > >
> > > > > > I would also like to give some feedback on a few aspects of the reviewing process that might help ease interactions with reviewers in the future. A few points to remember are that
> > > > > > 1) While reviewers are also dedicated to responding, they also have their own responsibilities. Reviewers may not be able to respond over the weekend. Hence, if faster responses are desired, consider posting responses before Friday.
> > > > > > 2) Reviewers can be anywhere in the world. Hence, even if it is 8am for one person, it could be 8pm for another. This may be felt especially before the weekends.
> > > > > > Overall, the posting times of authors greatly impact the response times of reviewers.
> > > > > >
> > > > > > 1/3) Thank you for clarifying your process.
> > > > > > 2/3) I do not agree with this line of reasoning. Just because the evaluation may never be perfect does not mean that we do not need to try. The fact is that the number and broadness of datasets presented in this paper fall far behind other literature in the field, as in the papers I pointed out in the original comment. As a field, we should not move away from the intended target, but rather towards it. If the datasets the paper is testing on show minimal differences to any method, then better datasets must be found that have more room for improvement and discuss why these work better (e.g. less dataset noise, fine-grained datasets are more successful, performance has simply saturated on some datasets, etc). As the work stands, the testing datasets show too marginal of differences anywhere for me to find them convincing. The addition of Aircraft helps, but the results are still far behind what I would consider convincing on a broader scale (once again, especially when compared to the experiments in comparable work). Maybe this method works very great, but the experiments have not shown this.
> > > > > > 3/3) These are all great explanations that should be in the paper in some way. It is simply common practice to include some ablations in a paper, to show some reasoning for why one method was chosen instead of something simpler or less expensive.
> > > > > >
> > > > > > As a whole, I chose to keep my original rating, as I feel confident that it fits. Overall, I believe the experimental section must be much stronger to merit publication in a venue like this. I wish you the best of luck.

---

> > > ### Comment · Reviewer_9gWJ · 2024-11-25
> > > **Clarification 2 / 3**
> > >
> > > Thank you for the in-depth explanation. Looking at your results again, I see your point that even small gains may be significant in this setting, considering that there is such a low improvement from e.g. vanilla to multi as well. However, I see this as an additional argument that the datasets need to be more fitting, as addressed in response 1 / 3. But to be fair, your new results on the new dataset seem to be more significantly better (especially EuroSAT). I also see your point about the cost coming more at training time, but less at inference.
> > >
> > > Overall, I really appreciate the additional explanation and results, and it improves my outlook on your paper. However, I am a bit unsure. The underlying point is that the additional time for DeCap is worthwhile, as long as the results are improving. So far, I am convinced very much by the results on EuroSAT and CIFAR10. I am somewhat convinced by the results on Aircraft. Overall, this just feels a bit thin for me to significantly raise my score.

---

> > > > ### Author Response · Authors · 2024-11-29
> > > > **Response to Clarification 2 / 3**
> > > >
> > > > ## datasets need to be more fitting
> > > >
> > > > 1.The performance of synthetic data indeed varies across different datasets, which is primarily due to the inherent characteristics of each dataset (e.g., the collectors’ understanding of class labels, such as whether an ostrich qualifies as a bird) , and more importantly, the generative model's capability. The differences in training distributions for generative models may result in subpar performance for certain objects. For example, they may good at “car” class, but are weak in “nematode” class (You can try this in https://clipdrop.co/instant-text-to-image). Therefore, while this issue exists, it is not unique to our method but rather a general challenge in the field.
> > > >
> > > > ## The underlying point is that the additional time for DeCap is worthwhile, as long as the results are improving.
> > > >
> > > > 2.1 As stated in 1, the differences is mainly due to datasets themselves. Some datasets benefit significantly, while others may not gain much from synthetic data. The same conclusion are also noted in [1]. Furthermore, current methods often exhibit inconsistent performance across datasets, whereas our approach consistently outperforms them, which is a substantial advancement. Additionally, since our method selects data from the prompt pool, if an existing method has already approached the optimal solution, our improvement may appear less significant. However, in real-world scenarios, these existing methods often vary greatly across datasets and cannot guarantee optimal performance. Thus, our method remains valuable in such contexts.
> > > >
> > > > 2.2 We may clarify that: The primary advantage of our method lies not in discovering a specific solving algorithm but in model formulation at the task level. As mentioned in Response 4.3.2, with further development in the generative modeling field, our framework can be seamlessly adapted to use gradient descent for optimization. In such a scenario, prompt optimization and lower-level model optimization can occur simultaneously. Compared to training-free methods, the additional time cost would merely involve a few hundred gradient descent steps with respect to $\theta$.
> > > >
> > > > 2.3 Besides, we want to kindly clarify that the results may not be improved through consuming more time. Existing methods, despite taking up so much time as ours, still will fail to achieve our results. To illustrate this, we show the results of the LE method generating different numbers of prompts on the STL10 dataset. We can see that simply enhancing prompt numbers may not bring classification performance gain, which we contribute this phenomenon to subpopulation shift as stated in Introduction section. This result also state the importance of proposed DeCap’s model in mining proper prompts.
> > > >
> > > > Prompt numbers per class| 100   | 200   | 300   | 400   | 500   |
> > > > |---------|-------|-------|-------|-------|-------|
> > > > STL10| 94.58 | 94.61 | 94.61 | 94.63 | 94.59 |
> > > >
> > > > [1].Is synthetic data from generative models ready for image recognition? ICLR,2023.

---

> > ### Comment · Reviewer_9gWJ · 2024-11-25
> > **Clarification 1 / 3**
> >
> > 4.1 I believe these additional datasets are very valuable, and would like to thank the authors for adding them. They have assuaged my concerns in this area as much as is possible at this time, considering the limited timeframe
> >
> > 4.2 I would like to clarify, are you generating the images at 512x512 and then downsampling them to the lower resolution, or generating them directly at the lower resolution? Because downsampling should not cause issues (to your point), but my concern is that if you're generating them directly at the lower resolution then the results would not generalize well to images generated at the in-distribution resolution. The training resolution should be 512x512, not 224x224, correct? Then the comparison would need to be done with the resolution at which the model was trained (in this case, 224 is already OOD and could suffer the same generalization problems as 96, e.g.)

---

> > > ### Author Response · Authors · 2024-11-29
> > > **Response to Clarification 1 / 3.**
> > >
> > > ## Are you generating the images at 512x512 and then downsampling them to the lower resolution, or generating them directly at the lower resolution?
> > > We apologize for not explaining this point clearly, which may have caused confusion. Our approach involves two steps: firstly generating images at a resolution of 512 and then downsampling them to the lower resolution. More specifically, since we use a black-box optimization algorithm for solving the outer-loop problem, we can directly convert the generative model outputs into the PIL.Image format for manipulation—potentially treating them as real images without any issues.
> > > As for why we do not generate low-resolution images directly, the reason lies in the current limitations of large generative models, which still struggle to generalize well across different resolutions. In other words, they perform best when generating images at the resolution they were trained on (in our case, 512, which aligns with what you mentioned). Therefor, in many guidelines for large generative models, the recommended resolution is often explicitly stated (e.g., https://huggingface.co/docs/diffusers/using-diffusers/sdxl_turbo).
> > > Your concern is valid, but the reality is often even more problematic—other resolutions may fail to generate usable images entirely. To provide a clearer understanding, we have included some simple examples in the Fig.13 of Appendix to illustrate this point.

---

> ### Author Response · Authors · 2024-12-02
> **Looking forward to your feedback**
>
> Dear Reviewer 9gWJ,
>
> The additional discussion period will come to an end. We kindly request your feedback on whether our new responses have effectively addressed your concerns.
> Best regards,
>
> The Authors

---

### Official Review · Reviewer_1Eaj · 2024-11-02

**Soundness:** 3
**Presentation:** 3
**Contribution:** 2
**Rating:** 6
**Confidence:** 5

**Summary:**

The paper proposes a Diversity-Enhanced and Class-Aware Prompt (DeCap) learning strategy to discover appropriate text prompts for downstream few-shot classification tasks, which addresses the issue of the lack of diversity in existing prompt-based methods. The proposed method includes a diversity-enhanced prompt pool construction and a class-aware prompt learning strategy. Empirical experiments are included to demonstrate the effectiveness of this approach.

**Strengths:**

1. This paper presents an innovative strategy that involves constructing more diverse prompts and introduces a prompt-learning technique tailored for downstream classification tasks to achieve optimal prompts. The approach is novel, and its effectiveness is supported by experimental results.
2. The paper systematically compares and summarizes various methods of data curation through prompts in T2I models. This overview is valuable for readers seeking to understand progress in this area.

**Weaknesses:**

1. Although this work proposes an interesting gradient-free method for discovering effective prompts for synthetic data, the main limitation is that “**attempting to approximate the few-shot data domain by altering prompts is inefficient and lacks generalization capability**”.

- **Regarding efficiency**: The meta-learning paradigm requires substantial sampling, and the pipeline includes too many uncontrolled components. For instance, generating images from text involves a high degree of freedom, while even slight variations in the images can significantly alter the classifier’s decision boundary. Thus a discussion of the running time should be included, along with an analysis of the strategy’s robustness with respect to T2I hyperparameters beyond prompts, such as generation using fixed seed to a random seed.
- **Regarding generalization**: The genetic algorithm essentially searches for effective prompts to approximate the domain of the few-shot dataset (assuming Equation 2 uses accuracy on the few-shot data as the fitness function for prompts rather than accuracy on the test set). Consequently, this approach may limit performance on out-of-domain test scenarios. Results on ImageNet domain generalization datasets, such as ImageNet-R, would be beneficial to verify the generalization capability.

2. Some recent data augmentation studies are not included [1,2,3]. Leveraging the inherent knowledge in few-shot images directly to curate data, bypassing text prompts, may offer a more efficient solution. Adding discussions and comparisons between prompt-based and image-based (e.g., image editing) data curation methods would provide better insight, such Real-Guidance [4] and Da-fusion [1].


3. The impact of varying shot numbers on the prompt-mining strategy is not addressed. Section 2.2 indicates that all experiments used a 10-shot setting. I suspect that the strategy may be ineffective in extremely low-shot cases (e.g., 1-shot). As noted in Weakness #1.2, this could result in synthetic data failing to adequately cover the evaluation dataset. Can you provide the more results across scenarios with varying shot numbers?

[1] DA-fusion: Effective Data Augmentation With Diffusion Models. Brandon Trabucco, et-al.

[2] Enhance Image Classification via Inter-Class Image Mixup with Diffusion Model. Zhicai Wang, et-al.

[3] Diffusemix: Label-preserving data augmentation with diffusion models. Khawar Islam, et-al.

[4] Is synthetic data from generative models ready for image recognition? Ruifei He, et-al

**Questions:**

My questions regarding efficiency and generalization capability are shown in the weakness.

---

> ### Author Response · Authors · 2024-11-24
> **Response Part 1/3**
>
> # About generalization and efficiency
> ## Response 3.1:
> - Firstly, we want to kindly clarify that our method does not attempt to approximate the few-shot data domain by altering prompts. Actually, most of existing methods are primarily limited to generating data by hand-designed prompts for approximating the few-shot data domain. These approaches are prone to overfitting and may generate images that are very similar to the few-shot data. To overcome the limitations, we explore to utilize the few-shot data to evaluate the quality of generated images at the higher task(meta)-level. In other word, we pay emphasis on the classification performance of the few-shot task rather than visual similarity of the few-shot data. This could be validated in Fig.6 of Appendix D.6, where the generated images using mined prompts by DeCap method have visually distinct characteristics from the real data, but the classification accuracy of model trained with only generated images closely matches to that trained with the full real dataset. Specifically, we trained our model using PACS-Sketch dataset, but the learned prompts encompass diverse styles, such as painting and cartoon, rather than being limited to visual similarity alone.[Additionally, we compare our method with previous methods that aim to approximate few-shot data domain (also see Response 3.3)] and these experiments clearly demonstrate the superiority of task-level data generation.
>
> - With such novel meta-learning formulation, we want to further clarify that our DeCap method is efficient and has good generalization capability. Actually, the meta-learning paradigm consists of two stage: meta-training and meta-test. Though it requires some cost to mine proper prompts during meta-training stage, the generalization and efficiency is reflected at the meta-test stage. Specifically, the generalization is reflected in the ability to transfer from the available few-shot dataset to the full dataset--just see Table 1 of Section 4.1, which we are only access to few-shot dataset during training, but successfully transferred to full dataset during testing.  And the efficiency is demonstrated by the fact that the learned prompts can be directly applied to meta-test tasks without further tuning. This means when we finish training, our method can enjoy the same speed as hand-crafted methods.
>
> In a nutshell, we admit that our method is still relatively time-consuming during the meta-training phase, while at the meta-test stage, our method could exhibits both efficiency and generalization. This is brought by the meta-learning formulation for the problem, rather than approximate the few-shot data domain by altering prompts.

---

> ### Author Response · Authors · 2024-11-24
> **Response Part 2/3**
>
> # Addtional questions related to generalization and efficiency
> ## Response 3.2:
>
> - Regarding efficiency: Firstly, we want to kindly clarify that although generating images from text involves a high degree of freedom, the classification boundaries of model trained on these generating images are not as fragile as one might expect in our experiments. On the contrary, they are exceptionally robust.
>
> To illustrate this, We provide two experiments:1) we give the classification performance on Imagenet100 dataset equipped with different generate numbers per class, in order to show classification boundaries are not sensitive to slight variations. The same finding is also found in [1]. 2) in Table 1 of our experiments, we already used different random seeds for testing and the results are quite stable (However, due to space constraints, we removed the standard deviation in the latest PDF. You can refer to earlier versions, or if you still have doubts, we can provide the data for your reference.). The results show that the model exhibits good robustness. These two points strongly support the feasibility of optimizing prompts for dataset construction.
>
> Additionally, for the issue of too many uncontrollable components, we controlled these components during the meta-training phase, including the initialization of the downstream model, random seeds for data generation, and data augmentation methods. However, during the meta-test phase, we did not control any of these components, yet our experimental results still show that our method is effective. This may attribute to the generalization capability of our approach. As for the random seed hyperparameters you mentioned, we have already incorporated them into our experiments. If you're interested in other hyperparameters, please feel free to ask.
>
> | Numbers per class | 100   | 200   | 400   | 600   | 800   | 1000  |
> |-------------------|-------|-------|-------|-------|-------|-------|
> | Imagenet100       | 70.42 | 70.47 | 70.52 | 70.62 | 70.64 | 70.62 |
>
> - Regarding generalization: To illustrate the out-of-distribution (OOD) generalization capablity of our DeCap method, we conduct the following experiment: we used CLIP's linear probe for training on the ImageNet100 dataset to eliminate any information inherent in CLIP's text encoder and then directly tested on various datasets without any further training. The results are shown below.  We attribute this phenomenon to the diversity-enhancing capability of DeCap.
>
> |                 | vanilla prompt | multi-domain | LE    | CiP   | DeCap (ours)   |
> |-----------------|----------------|--------------|-------|-------|----------------|
> | ImageNet-Sketch | 7.81           | 23.42 | 11.2  | 9.6   | 22.13          |
> | ImageNet-v2     | 61.07          | 66.53        | 60.73 | 67.9  | 70.43 |
> | ImageNet-C      | 23.84          | 29.45        | 29.55 | 24.89 | 33.51 |
> | average         | 30.9           | 39.8         | 32.59 | 34.97 | 42.02 |
>
> [1].Fake it till you make it: Learning transferable representations from synthetic ImageNet clones.CVPR,2023.

---

> > ### Comment · Reviewer_1Eaj · 2024-12-01
> > **Clarification 1 / 2**
> >
> > Thank you for the additional discussion regarding efficiency and generalization. I find the arguments for robustness and OOD generalization using heuristic prompt design convincing. I also agree with the authors’ clarification that the method is not merely overfitting to the few-shot domain. DeCap does not explicitly learn the few-shot visual domain, which helps explain why this approach is likely to achieve better generalization.

---

> ### Author Response · Authors · 2024-11-24
> **Response Part 3/3**
>
> # Compared with image-based data curation methods
> ## Response 3.3:
> Thanks for your valuable suggestions for the related papers, we have cited them in our manuscript. And we  clarify differences and connections with our method as follows.
>
> - Current research on prompt-based data generation methods mainly focuses on how to generate images when data are scarce or unavailable for the concerned task, while data-based data augmentation methods are more focused on how to further boost the performance when data are relatively enough. We can see that both of them have their own advantages on specific problems, and they could be used simultaneously as done in the papers given by the reviewer. In this paper, we focus on prompt-based data generation methods, and thus we compare the prompt-based data generation aspects of these works for a fair comparion. These prompt-based data generation methods can be broadly categorized into two types: textual inversion for learning prompts and the use of SDEdit technology to generate images. We provide the compared results using SDEdit, textual inversion, and DeCap on the STL10 and Caltech-101 datasets as follows:
>
> | Methods     | SDEidt | Textual inversion | DeCap |
> |-------------|--------|:-------------------:|-------|
> | STL10       | 94.74  | 94.86 | 95.9  |
> | Caltech-101 | 85.82  | 84.76 | 88.67 |
>
> Above results show that our method has an consistent advantage over these methods under few-shot problem (i.e., scarce data). Of course, image-based data augmentation methods can be applied to further boost our approach and compared baselines. This would be explored in our future work.
>
> - Here, we explore to use image-based data generation methods to alleviate the issue of the downstream few-shot tasks with extremely scarce data. Specifically, one could use image-based data augmentation techniques to create an augmented real dataset and then apply our method to mine proper prompts for generating high-quality images, and the experimental results are described in the response 3.4.
>
> # Improved strategy in extremely low-shot cases.
> ## Response 3.4:
> Thanks for this valuable suggestion. We addtionally provide the results for the 1-shot scenario on the STL10 dataset. As it shown, even under extremely scarce data, our method could achieve better performance than CiP method with 10-shot images. By using image-based data generation methods to alleviate the issue of scarce data, the performance of our method could be further increased.
> However, we would admit that, due to the limitations of scarce data in characterize few-shot task, hand-crafted prompt methods (the vanilla prompt) could yield better results by utilizing the knowledge embedded in the generative model. When given data increase to 10-shot, our method could achieve competitive results. And we will explore to better address the issue in the future work.
>
> |        | 1shot | 1shot+aug | 10shot | vanilla | CiP   |
> |--------|-------|:-----------:|--------|---------|-------|
> | result | 95.15 | 95.2      | 95.9   | 95.33   | 94.92 |

---

> > ### Comment · Reviewer_1Eaj · 2024-12-01
> > **Clarification 2/2**
> >
> > Thank you for your additional comparison with image-based data curation methods. I agree that, in data-scarce settings, image-based augmentation tends to be less effective due to limited diversity. However, this reduced diversity can be mitigated by introducing prompt engineering during editing (e.g., using a different class prompt when conducting SDEdit). Therefore, I believe that combining image-based curation with prompt-based curation holds significant potential as a more effective data curation strategy.
> >
> > For this work, which focuses solely on prompt-based methods, I acknowledge the efforts made on the prompt side. Considering the robustness discussed in Response 2/3 and the image-based comparison in Response 3/3, I now hold a more positive view.

---

### Official Review · Reviewer_W3vA · 2024-11-03

**Soundness:** 2
**Presentation:** 1
**Contribution:** 1
**Rating:** 3
**Confidence:** 5

**Summary:**

The authors propose a prompt learning method to boost classification performance. To overcome the lack of diversity of hand-crafted prompts, they also leverage prompts from language models. Once a prompt pool has been created, genetic algorithm finds out the best working prompt for classification performance.

**Strengths:**

- The method is easy to follow.
- It proposes a total solution for few-shot classification problem.

**Weaknesses:**

- At the first glance of this paper, the though 'commercially or open-sourced LLM or Multi-modal LLM will show much better performance with even not using genetic algorithm' came up in my mind. That means many readers would think just the same I did and the authors are responsible for prove them wrong. I have tried myself requesting various prompts to a commercial LLM and the results seemed much better than the examples in this paper.
- The task is limited to classification problem and datasets used for evaluation are insufficient to make this paper solid. It is hard to tell if this paper worth dropping another paper for this conference.

**Questions:**

Comments in weaknesses must be resolved.

---

> ### Author Response · Authors · 2024-11-24
> **Response Part 1/2**
>
> # Commercially or open-sourced LLM or Multi-modal LLM will show much better performance with even not using genetic algorithm.
> ## Response 2.1:
> We want to kindly clarify that our work is different from existing commercially or open-sourced LLM or Multi-modal LLM.
>
> - Firstly, existing commercially or open-sourced LLM or Multi-modal LLM focused on improving quality of generated images, while they pay less attention to using generated images to solve downstream tasks. As a result, although their capabilities in improving image generation are becoming better, e.g., high-resolution images, this does not mean that they could generate images that are more helpful for downstream tasks, e.g., few-shot classification tasks investigated in this paper. Previous works like [1-7] have explored to use open-sourced Stable Diffusion models to help generate images for classification task. However, existing works are relatively hard to guarantee that training models from synthetic images are efficient for downstream classification tasks. To further improve the quality of generated images for classification, we propose a novel prompt learning strategy for improve downstream few-shot classification tasks.
>
> - Secondly, the latest commercially or open-sourced large language models could generate text prompts for text-to-image generative models to generate high-quality images. However, no matter how strong the generation ability or how high the text quality is, the generated text prompts are always less related to downstream tasks. Some generated text may be helpful for training, but others may have no or even negative impact on classification performance (Please also see Figure 3 of the manuscript). To address this issue, we propose a novel meta-learning approach to adaptively mine proper prompts tailored for the few-shot learning task. Our experimental results further support that mined prompts are attained specifically suitable to concerned few-shot learning task. Especially, our method has weak correlation to open-sourced large language models.
>
> - Thirdly, we want to kindly clarify that the powerful large language model is not always better. Current text-to-image generation models may not fully understand complex text description, which may lead to overly detailed prompts by LLM may perform poorly. As compared, our adaptive prompt learing strategy could learn proper prompts tailored for the few-shot learning task.
>
> - Fourthly, we additionally provide results on EuroSAT and FGVC Aircraft datasets to illustration the superiority of proposed method. The results are shown in Table 1,2 of Section 4.1 of the manuscript. These datasets present more significant challenges than commonly used datasets. We use text prompts generated by ChatGPT to construct the prompt pool, and then use our method to select the proper propmts. The results show that our method has a significant advantage over simply use ChatGPT to generate prompts. This strongly supports our viewpoint.
>
> In a nutshell, we believe our work would inspire more researchers in the community to focus on the limitations of current LLMs, and explore more meaningful researches built upon the advancements of LLMs.
>
> [1].Synthetic Data from Diffusion Models Improves ImageNet Classification. TMLR,2023.
>
> [2].SuS-X: Training-Free Name-Only Transfer of Vision-Language Models. ICCV,2023.
>
> [3].Prompt, Generate, then Cache: Cascade of Foundation Models makes Strong Few-shot Learner. CVPR,2023.
>
> [4].Is synthetic data from generative models ready for image recognition? ICLR,2023.
>
> [5].Fake it till you make it: Learning transferable representations from synthetic ImageNet clones. CVPR,2023.
>
> [6].Leaving Reality to Imagination: Robust Classification via Generated Datasets.ICLR,2023.
>
> [7].DA-fusion: Effective Data Augmentation With Diffusion Models. ICLR,2024.

---

> ### Author Response · Authors · 2024-11-24
> **Response Part 2/2**
>
> # The task is limited to classification problem and datasets used for evaluation are insufficient to make this paper solid.
> ## Response 2.2:
> Thank you for recognizing the versatility of our method. As a framework, we also believe that DeCap has great flexibility and potential for extension to other domains, but this is not an easy task. In fact, our algorithm incorporates several unique designs specifically for classification that enable it to achieve classification-awareness.
>
> For other tasks, such as object detection, how to correctly utilize the labeled information from generated data—such as understanding which pixels the text encoding is sensitive to—becomes the biggest challenge. This is not trivial, and we discuss the limitations of our method in detail in Appendix A, hoping that our paper can inspire further research in this area and contribute to other domains.
>
> Additionally, many of the current works[1-7] on generating datasets focus on classification tasks. Extending their research ideas in this way does not seem too narrow. Lastly, our dataset already includes categories such as objects, remote sensing, fine-grained, low resolution, and specific domains, and we are preparing a scene dataset for training. If you feel that our dataset is still insufficient, we would appreciate more detailed suggestions.
>
> type     | datasets
> -------- | -----
> objects  | STL10, CIFAR10, Im-10, Caltech-101, Im-100
> remote sensing  | EuroSAT
> fine-grained  | Pets, Aircraft
> low resolution  | CIFAR10, EuroSAT
> specific domain  | PACS-sketch
>
> [1].Synthetic Data from Diffusion Models Improves ImageNet Classification. TMLR,2023.
>
> [2].SuS-X: Training-Free Name-Only Transfer of Vision-Language Models. ICCV,2023.
>
> [3].Prompt, Generate, then Cache: Cascade of Foundation Models makes Strong Few-shot Learner. CVPR,2023.
>
> [4].Is synthetic data from generative models ready for image recognition? ICLR,2023.
>
> [5].Fake it till you make it: Learning transferable representations from synthetic ImageNet clones. CVPR,2023.
>
> [6].Leaving Reality to Imagination: Robust Classification via Generated Datasets.ICLR,2023.
>
> [7].DA-fusion: Effective Data Augmentation With Diffusion Models. ICLR,2024.

---

> ### Author Response · Authors · 2024-11-29
> **Response to your further problem. Part 1/2**
>
> ## The authors are arguing that exploiting given prompts will always show better performance. However to do so, there should be evidence about that hypothesis.
> R: In fact, we do not assume that prompts will necessarily lead to better results; many prompts can even produce worse outcomes for downstream classification tasks. This is elaborated in detail in Figure 3 of the manuscript. For example, in the cat-and-dog classification task, prompts containing references to both cats and dogs (e.g., "a dog plays with a dog") can negatively impact classification performance.
> These results show it is challenging to directly exploit given prompts to improve downstream classification performance. To address this limitation, we propose to mine proper prompts from given prompt pool which are attained specifically suitable to concerned downstream classification task. The result in Table 1, 7, 8, and 9 support the capability of proposed method in extracting effective prompts, demonstrating that our approach achieves better results across various metrics and datasets compared to existing algorithms.
>
> ## It is hard to say that a Genetic algorithm applied upon training a classification model is simple and practical.
> R: Other reviewers acknowledged the effectiveness and practicality of our method. Our experimental results (Tables 1, 2, 7, 8, and 9) demonstrate this point across classification performance, generalization, and various evaluation metrics. As for simplicity, we want to kindly clarify that our method is relatively time-consuming during the meta-training phase, while is high efficiency and generalization during the meta-test stage. This point is also acknowledged by other reviewers.
>
> Specifically, we would like to restate our method to provide a clear understanding of its generalizability and cost. Existing data generation approaches are relatively limited to approximations few-shot data domain at the data level, and thus the generating images tend to be similar to given few-shot data, which may lead to overfitting (we provide some results compared with common “approximating few-shot data” methods as follows). To address this issue, we attempt to rethink the way of utilizing few-shot data. In our implementation, we move beyond just visual similarity and focus on the task-level utilization of few-shot data to improve downstream classification performance. As validated in Fig.6 of Appendix D.6, the prompts we learned would generate images with significantly different visual features from the real few-shot data. While the classification accuracy of models trained with these synthetic data approaches that of models trained on the full dataset. Furthermore, we compare our method with previous data generation methods in Tables 1, 2, 7, 8, and 9, and the experimental results clearly show the superiority of our method in improving data generation for downstream classification performance.
>
> | Methods     | SDEidt | Textual inversion | DeCap |
> |-------------|--------|:-------------------:|-------|
> | STL10       | 94.74  | 94.86             | 95.9  |
> | Caltech-101 | 85.82  | 84.76             | 88.67 |
>
> To achieve this goal, meta-learning is naturally employed to provide task-level guidance for generating classification-aware images. We achieve an adaptive sample selection process that links generated data with downstream classification task. In meta-learning community, the generalization and efficiency are emphasized during the meta-test phase. In our case, generalization is demonstrated by the ability to scale from accessible few-shot datasets to full datasets, and efficiency is reflected in the fact that the learned prompts can be directly applied for generalization without any further tuning. This means when we finish meta-training process, our method can enjoy the same speed as previous hand-crafted prompt methods. Therefore, during the meta-test phase, our method tends to demonstrate high efficiency and generalization. Of course, we also acknowledge that our method is still relatively time-consuming during the meta-training phase, as discussed in Appendix A (limitations). However, the advantage of data generation for improving downstream classification task is promising to overcome the limitation of existing methods.
>
> Besides, genetic algorithm is currently an alternative to gradient backpropagation, and has already proven the feasibility of the proposed approach. With further advancements in large-scale generative models and their using cost decreasing, our framework can be easily transformed into a white-box optimization model, which will significantly reduce runtime. Currently, we have to retrain the inner loop each time, but with white-box optimization, we can continue training with the weights from the previous loop. By that time, the time cost will no longer be an issue.

---

> > ### Author Response · Authors · 2024-11-29
> > **Response to your further problem. Part 2/2**
> >
> > ## LLMs nowadays show remarkable performance on prompt generation and if the amount of the data is plenty enough, I think the performance gain would be cost-effective.
> >
> > R: When the number of prompts is fixed, increasing the number of generated images does not yield significant benefits. This has been validated by [1], which generated a dataset 50 times larger than ImageNet but found the performance improvement to be marginal and not cost-effective.
> >
> > Increasing the number of prompts also does not bring as much benefit as one might expect, It’s also shown in [1]. In fact, in our experiments, the number of prompts generated by the LLM-based method (referred to as the LE method) far exceeds that of proposed DeCap method. For example, on datasets such as Caltech-101, LE generated over 10,000 prompts, while DeCap only mines only about 500 prompts. However, proposed DeCap method could achieved superior results with 20 times fewer prompts. Moreover, to further state this point, we give extra experiments about the influence of different LE prompt numbers on the STL10 dataset. Simply enhancing prompt numbers may not bring classification performance gain, which we contribute this phenomenon to subpopulation shift as stated in Introduction section.
> >
> > Prompt numbers per class| 100   | 200   | 300   | 400   | 500  |
> > |-------|-------|-------|-------|-------|-------|
> > STL10| 94.58 | 94.61 | 94.61 | 94.63 | 94.59 |
> >
> > [1].Fake it till you make it: Learning transferable representations from synthetic ImageNet clones. CVPR,2023.
> >
> > [2].Is synthetic data from generative models ready for image recognition? ICLR,2023.

---

> > > ### Comment · Reviewer_W3vA · 2024-12-03
> > >
> > > The authors do not seem to get my point and I think it would be hard to meet in the middle. I summarize my unresolved comments as below.
> > >
> > > - I still think there should be a presentation of 'Generating from well designed prompts vs picking out best prompts out of roughly designed prompts'.
> > > - Assuming myself as a potential user of this method, the experimental settings are not solid enough. The resolution and amount of the data is not large enough while using genetic algorithm is heavy to use.
> > > - Whether the amount of data does not affect the performance relies on the problem setting. 94% of precision shows that the problem itself is not challenging enough and the error is likely to be easily affected by the artifact in test set.
> > >
> > > Combining all these comments, if the authors have chosen more generic and large dataset, proven failure of LLMs available in the community, and still made an undeniable performance enhancement, I would be open to give an accept to this paper.
> > > By now, I will keep the score as-is.

---

### Official Review · Reviewer_Bqtv · 2024-11-04

**Soundness:** 3
**Presentation:** 3
**Contribution:** 3
**Rating:** 6
**Confidence:** 4

**Summary:**

The paper presents DeCap, a novel method for enhancing few-shot learning through the generation of diverse and classification-aware synthetic images using text-to-image generative models.
The proposed automated prompt learning method reduces the manual effort required to craft effective prompts and its meta-learning strategy can align synthetic data generation with downstream classification tasks to improve the performance in various network architectures.

**Strengths:**

The innovation of this work lies in the adaptive learning of prompts for synthetic data generation, which is tailored to the specific requirements of few-shot learning tasks.
This work use a genetic algorithm for the outer-loop optimization to search for the optimal prompt set, which is a discrete search problem. The inner-loop optimization involves training a classification model on synthetic data generated by the selected prompts.
The paper demonstrates that DeCap can improve the performance of existing zero/few-shot learning methods in different network architectures

**Weaknesses:**

A significant concern with DeCap's approach is the adequacy of the prompt pool in capturing the full spectrum of variations necessary for diverse few-shot learning tasks. The paper does not provide a clear methodology for ensuring that the prompt pool is comprehensive enough, or that the most contributive prompts are consistently selected across different optimization iterations and datasets. This lack of clarity raises questions about the reliability and robustness of the prompt selection process.

The paper primarily relies on accuracy as the performance metric for evaluating the few-shot learning models. However, in the context of few-shot learning, a more nuanced evaluation that includes "n-shot n-way" scenarios and a suite of metrics such as precision, recall, and F1 score is essential. The current approach may not fully capture the model's performance across different classification tasks, especially when the distribution of samples per class varies significantly. This limitation restricts the comprehensive assessment of the model's robustness and generalization capabilities in few-shot settings.

**Questions:**

Could you provide a few examples of prompts that were either consistently selected or commonly rejected across different datasets? Additionally, would it be possible to include visualizations that demonstrate the prompt selection process over the course of your optimization? Such examples and visualizations would greatly aid in understanding the behavior of your DeCap method and the rationale of the prompts it selects for synthetic image generation？

A critical aspect of DeCap's approach is the ability to generate diverse and classification-aware synthetic images, which hinges on the richness and detail of the prompt pool. My concern is whether the current prompt pool contains a sufficient level of fine-grained detail to effectively capture the diversity required for various few-shot learning tasks. Could you elaborate on how you ensure that the prompts in your pool are not only comprehensive but also detailed enough to represent the nuances of different classification tasks? Additionally, how do you assess whether the prompts selected across different optimization iterations and datasets are indeed the most contributive ones in terms of fine-grained detail?

Could you consider expanding the evaluation framework to include a range of "n-shot n-way" scenarios? The paper would benefit from a more holistic evaluation that includes the F1 score, precision, and recall in your analysis. Specifically, how would these metrics help in understanding the model's generalization performance in terms of false positives and false negatives, which are critical considerations in few-shot learning?

---

> ### Author Response · Authors · 2024-11-24
> **Response Part 1/2**
>
> # The comprehensiveness of propmt pool
> ## Response 1.1:
> In our formulation, we aim to identify an appropriate prompt set $\theta$ from the prompt pool $\Theta$ (a approximate construction of the prompt hypothesis space). Ideally, $\Theta$ contains a comprehensive set of potential prompts related to the concerned tasks. Previous methods often design prompts focusing on specific aspects. For example, hand-crafted prompts are tailored for different domains (please see Figure 1(a)), while model-generated prompts may contain rich content information (Please see Figure 1(b)). In contrast, our prompt pool construction aims to incorporate information from diverse perspectives. To enhance domain diversity, we manually curate domain-specific information. To enrich the content diversity of prompts, we utilize language enhancement techniques to generate a large number of diverse prompts. To improve task relevance, we adopt the CiP method to capture prompts closely related to the task images. Consequently, our approach significantly improves comprehensiveness of prompts compared to existing methods.
>
> However, we admit that creating a comprehensive enough prompt pool is challenging because the optimal prompt cannot be guaranteed to exist within the constructed pool (as the prompt hypothesis space is theoretically continuous). Nonetheless, our experiments demonstrate that the constructed prompt pool sufficiently meets the needs of downstream classification tasks. From the given pool, the proposed algorithm adaptively learns the proper set of prompts, leading to performance improvements. Notably, although the constructed prompt pool is discrete, the learned optimal prompts exhibit stronger interpretability compared to those in a continuous hypothesis space. Detailed examples are provided in Fig.3 of Section 4.3 and Fig.10,11,12 of Appendix E.
>
>
> # About prompt selection process
> ## Response 1.2:
> - Though our method adaptively learns proper prompts from the data, the learned prompts may share a consistent pattern. Specifically, the mined prompts almost consists of two parts: hand-crafted prompts and model-generated prompts.
>
> - Moreover, for specific categories, the ratio and content of model-generated prompts and hand-crafted prompts behaves differently. This demonstrates that our method can adaptively learn task-specific prompts for different datasets. We discuss these cases in detail in Fig 10,11 Appendix E.2.
>
> - In Table 4,5,6 of Appendix D.2, we demonstrate the improvement of our method compared to random selection and full selection. The experimental results show that our prompt selection method is relatively more reliable, as it leads to significant performance improvements. Moreover, in Fig.6 of Appendix D.6, we explore the impact of using only domain prompts in the prompt pool. As it shown, even when the prompt pool is not comprehensive, our DeCap method could still achieve performance gains, which strongly supports the robustness of our method. All of these illustrate that our method is potentially useful for diverse few-shot learning tasks.
>
> - While identifying prompts selection process across all datasets is challenging—since the satisfied prompts often vary depending on the specific dataset. However, the prompts selection/rejection process may emerge a general trend. On the one hand, all datasets tend to favor diverse domain prompts, such as "cartoon," "sketch," and "close-up," rather than being limited to the most common domains like "photo". On the other hand, prompts that are commonly rejected are shown in Fig.3 of Section 4.3. In summary, existing methods often face issues such as noisy class errors (e.g., both cat and dog appearing in a single image in a cat-dog classification task), model caption errors (e.g., identifying a monkey as a cat), and low-quality prompts (e.g., generating images that do not significantly aid classification). In contrast, our method rejects these prompts during the selection process.
>
> - Since our method focuses on the impact of generated data on downstream tasks, we only show classification performance and have not provided extensive visual demonstrations of prompt selection process. As suggested by the reviewer, we provide an example in Fig.12 of Appendix E.2 showing the evolution process of prompt selection for the airplane class in the STL10 dataset during iterative optimization. As it shown, there exist some improper prompt in the early stage, while more proper prompts are gradually selected during the iteration process, showing considerable diversity and encompassing multiple methods. Finally, the prompt selection process converges, and our method could mined proper prompt set for the concerned tasks.
>
> We believe above ‌interpretations of our prompt selection process could provide deeper understanding about our method.

---

> > ### Comment · Reviewer_Bqtv · 2024-11-26
> > **Re. Rebuttal**
> >
> > The author's emphasis on a comprehensive prompt pool is evident in their responses. However, to optimize synthetic image generation, it would be beneficial for them to highlight the significance of integrating fine-grained prompt descriptions. This could have the potential to mitigate issues arising from current prompting methods, such as low quality prompts. The author mentions the comparison of their prompt selection method with random and full selection in Tables 4, 5, 6 of Appendix D.2. However, the significant improvements or advantages of their method over random or full selection might not be clearly evident from these tables.

---

> > > ### Author Response · Authors · 2024-11-27
> > > **Some confusion**
> > >
> > > Thank you for your sincere suggestions. We will carefully revise our work based on your suggestions. However, we are slightly confused about the following points:
> > > - Could you clarify what you mean by “it would be beneficial for them to highlight the significance of integrating fine-grained prompt descriptions”? Does fine-grained prompt descriptions refer to the prompts generated by the model? In Table 11, we have newly added visualizations of consistently selected prompts generated by the model. However, if our understanding differs from your intended meaning, could you please provide a more detailed explanation?
> > > - Regarding your concern that “the significant improvements or advantages of their method over random or full selection might not be clearly evident from these tables,” we have already included results showing the performance of randomly replacing half of the prompts on each dataset to demonstrate the contribution of the selected prompts. Additionally, we are currently conducting experiments on random and full prompt selection across different datasets. Would you think these experiments are comprehensive enough? Is there anything else we should supplement?

---

> ### Author Response · Authors · 2024-11-24
> **Response Part 2/2**
>
> # About the assess of contributive prompts
> ## Response 1.3:
> To further illustrate whether or not the most contributive prompts are consistently selected, we conduct an ablation study: we randomly substitute half of the prompts DeCap learned with other prompts. The experimental results are presented as follows. It could be seen that when other prompts are substituted in learned prompts, the results show performance reduction. That is to say, the selected prompts indeed bring better classification performance on the concerned few-shot tasks.
>
> |            | STL10 | CIFAR10 | Im-10 | Pets  | Caltech-101 | Imagenet100 |
> |------------|-------|---------|-------|-------|-------------|-------------|
> | Substitute | 95.15 | 75.58   | 97.86 | 84.93 | 86.87       | 70.66       |
> | Origin     | 95.9  | 76.98   | 97.95 | 85.36 | 88.67       | 71.08       |
> # Why don’t we use the "n-way k-shot" setting. Some other metrics for showing model’s robustness and generalization.
> ## Response 1.4:
> In the field of few-shot learning, with the emergence of foundational models like advancements in data generation techniques, the community has shifted toward directly utilizing the given few-shot datasets to improve the concerned tasks, without access to any other data as done in ``n-shot n-way" scenarios, e.g., [1-5]. We just follow previous benchmarks in our study. However, your suggestion is extremely helpful, as we indeed overlooked the computation of additional evaluation metrics. As suggested by the reviewer, we provide the results of various metrics for each dataset (precision, recall, F1-score), as shown in Table 7,8,9 of Appendix D.3 of the manuscript. The results demonstrate that our method performs well on these metrics, indicating that it not only achieves high accuracy but also excels in identifying positive samples and is more cautious when dealing with them. That is to say, our method could generate a high-quality dataset to train a robust and generalizable model for concerned few-shot tasks.
>
> This capability is reflected not only in overall classification accuracy but also in precision, recall, and F1-score.
>
> [1].DA-fusion: Effective Data Augmentation With Diffusion Models. ICLR,2024.
>
> [2].SuS-X: Training-Free Name-Only Transfer of Vision-Language Models. ICCV,2023.
>
> [3].Prompt, Generate, then Cache: Cascade of Foundation Models makes Strong Few-shot Learner. CVPR,2023.
>
> [4].Is synthetic data from generative models ready for image recognition? ICLR,2023.
>
> [5].Generating images of rare concepts using pre-trained diffusion models. AAAI,2023.

---

### Author Response · Authors · 2024-11-25
**Common Response**

We thank all reviewers for their valuable feedback on our paper. We are happy to see that:

- Proposed adaptive prompt learning strategy tailored for downstream few-shot classification tasks was received as **innovative, novel, effective, valuable** by the reviewers;

- The paper is **well written and easy to follow**;

- The paper **systematically compares and summarizes** various methods of data curation through prompts in T2I models, and present a  **good show of qualitative results** to demonstrate the effectiveness of proposed approach;

convinced the reviewers as strengths.

While we acknowledge that our manuscript could be further improved. We have addressed all reviewer comments, including:

- requests for additional experiments on a diverse variety of datasets (Reviewer 9gWJ,Reviewer W3vA) - displayed in the PDF;

- questions about our methodology, e.g., the comprehensiveness of propmt pool and the visualizations of the prompt selection process (Reviewer Bqtv), generalization and efficiency (Reviewer 1Eaj);

- questions about our experiment setup, e.g., evaluation framework and metric (Reviewer Bqtv)

We hope we have addressed all concerns and look forward to the discussion phase.

---

### Meta-Review · Area_Chair_GQZA · 2024-12-17

**Metareview:**

This paper aims to enhance the performance of few shot learning by generating diverse and useful synthetic data from text-to-image models. The paper was reviewed by four knowledgeable referees who acknowledged that the paper is well written and easy to follow (W3vA, 9gWJ), the presented methodology is novel (1Eaj, 9gWJ), and the qualitative results are sufficient (9gWJ). The main concerns raised by the reviewers were:
1. Unconvincing experimental validation of the approach: missing metrics and datasets (Bqtv, 1Eaj, 9gWJ), limited scope (W3vA), missing comparisons with baselines / prior art (W3vA, 1Eaj, 9gW), missing ablations (9gWJ), and debatable significance/robustness of the presented results (Bqtv, 9gWJ)
2. Missing cost vs. improvements analyses (9gWJ)
3. Unclear efficiency and generalizability of the method (1Eaj)
4. Unclear adequacy of the prompt pool in capturing sufficient diversity (Bqtv)

The authors partially addressed the reviewers concerns during rebuttal and discussion. In particular, the authors included additional metrics (e.g. precision, recall, F1-score) and datasets, discussed the comprehensiveness of the prompt pool and selection process, and shared additional experimental evidence on OOD generalization, extreme few shot scenarios, and comparisons with other synthetic datasets. After rebuttal / discussions, reviewers remained unconvinced and kept raising concerns about the evaluation of the proposed approach. In particular, the improvements / advantages of the proposed approach are not convincing. The breadth of datasets considered for quantitative evaluation appeared to be narrower than the ones typically used in the literature, the differences appeared overall marginal making in challenging to assess the significance of the work, MLLM baselines would strengthen the contribution, and systematic ablations would help justify the proposed approach. Therefore, the result of the reviewers discussion period leans towards rejection. The MR agrees with the reviewers' concerns and therefore recommends to reject. The MR encourages the authors to consider the feedback received to improve future iterations of their work.

**Additional Comments On Reviewer Discussion:**

See above.

---

### Decision · Program_Chairs · 2025-01-22

Reject